

# Aerodynamic roughness length of crevassed tidewater glaciers from UAV mapping

Armin Dachauer[1,2], Richard Hann[3], and Andrew J. Hodson[2,4]

[1]Swiss Federal Institute of Technology in Zurich (ETH), Zurich, Switzerland
[2]The University Centre in Svalbard (UNIS), Longyearbyen, Svalbard
[3]Norwegian University of Science and Technology (NTNU), Trondheim, Norway
[4]Western Norway University of Applied Sciences, Sogndal (HVL), Norway

**Correspondence:** Armin Dachauer (armind@student.ethz.ch) and Richard Hann (richard.hann@ntnu.no)

**Abstract.** The aerodynamic roughness length ($z_0$) is an important parameter in the bulk approach for calculating turbulent fluxes and their contribution to ice melt. However, for heavily crevassed tidewater glaciers $z_0$ estimations are rare or only generalized. This study used unmanned aerial vehicles (UAVs) to map inaccessible tidewater glacier front areas. The high-resolution images were used in a structure-from-motion photogrammetry approach to build digital elevation models (DEMs).

These DEMs were applied to five different models (split across transect and raster methods) to estimate $z_0$ values of the mapped area. The results point out that the range of $z_0$ values across a glacier is large, with up to three (locally even four) orders of magnitude. The division of the mapped area into sub-grids (50 m $\times$ 50 m), each producing one $z_0$ value, best accounts for the high spatial variability of $z_0$ across the glacier. The $z_0$ estimations from the transect method are in general higher (up to one order of magnitude) than the raster method estimations. Furthermore, wind direction (values parallel to the ice flow direction are larger than perpendicular) and the chosen sub-grid size turned out to have a large impact on the $z_0$ values, again presenting

a range of up to one order of magnitude each. On average, $z_0$ values between 0.08 m and 0.88 m for a down-glacier wind direction were found. The UAV approach proved to be an ideal tool to provide distributed $z_0$ estimations of crevassed glaciers, which can be incorporated by models to improve the prediction of turbulent heat fluxes and ice melt rates.

## 1   Introduction

The aerodynamic roughness of a glacier influences the turbulent heat exchange between the glacier surface and the atmosphere (Rees and Arnold, 2006). Both sensible and latent heat fluxes balance this heat exchange on the surface and therefore have a large impact on the meltwater production and the surface energy balance of glaciers (Hock, 2005). The bulk aerodynamic method is a very popular approach for the calculation of those turbulent fluxes due to its low requirements for data collection. It only requires basic atmospheric measurements (e.g. wind speed, temperature) as well as the aerodynamic roughness length

of the surface (Chambers et al., 2020). The aerodynamic roughness length, also called $z_0$, is a length scale that represents the height above the surface at which the wind speed drops to zero (Chappell and Heritage, 2007). It is a constant surface characteristics (Lettau, 1969) describing the loss of wind momentum that can be attributed to surface roughness (Smith, 2014).



**Table 1.** Published aerodynamic roughness length values for crevassed glacier areas.

| Study | Method | Surface Type | $z_0$ (m) |
|---|---|---|---|
| Fitzpatrick et al. (2019) | Raster | Large crevasses | 0.01-0.5 |
| Obleitner (2000) | Transect | Very rough glacier ice | 0.05 |
| Smeets et al. (1999) | Transect | Very rough glacier ice | 0.02-0.08 |
| Smith et al. (2016) | Transect | Deep crevasses | 0.005-0.05 |
| Smith et al. (2016) | Raster | Deep crevasses | 0.003-0.025 |

Recently, a series of studies (e.g. Irvine-Fynn et al., 2014; Miles et al., 2017; Smith et al., 2016) were following the bulk approach to determine $z_0$ values of glacier surfaces while using digital elevation models (DEMs) as data source for their calculation. Terrestrial Light Detection and Ranging (LiDAR) systems (used for instance in the studies of Smith et al. (2016), Nicholson et al. (2016) or Nield et al. (2013a)) constitute a powerful way to effectively produce such DEMs. However, they are very expensive (Uysal et al., 2015) and limited in the area they cover (Irvine-Fynn et al., 2014). Thus, unmanned aerial vehicles (UAVs) provide a cheap alternative to overcome these limitations (Uysal et al., 2015). In recent years, UAVs have presented new opportunities for detailed mapping of earth surface and became more and more popular in the field of glaciology (Bhardwaj et al., 2016). The main advantage of UAVs is the possibility of collecting high temporal and spatial resolution data at low costs (Casella and Franzini, 2016) and to overcome the gap between sparse field observations and coarse resolution space-born remote sensing data (Bhardwaj et al., 2016).

Several studies already investigated the $z_0$ values of non-crevassed (Irvine-Fynn et al., 2014; Nield et al., 2013a), debris-covered (Quincey et al., 2017) or sparsely crevassed (Smith et al., 2016) ice surfaces using different approaches. However, still little is known about the effect of heavily crevassed glacier surfaces on the turbulent heat exchange between glaciers and the atmosphere. The greatly varying size of obstacles (i.e. crevasses) and the broader-scale, heterogeneous surface topography of such crevassed ice makes it hard to define $z_0$ values (Quincey et al., 2017). Typical values of $z_0$ on glacier ice range from less than 0.0001 m for smooth ice and 0.02 m to 0.08 m for rough glacier ice (Brock et al., 2006; Smeets et al., 1999). But published $z_0$ literature values for large and deep crevasses are rare. Table 1 gives a closer overview of such published aerodynamic roughness values and shows $z_0$ values up to 0.5 m for large crevasses (Fitzpatrick et al., 2019). Furthermore, most surface energy balance models only consider one single $z_0$ value for the whole glacier (commonly 0.001 m (Smith et al., 2020)), regardless of any spatial and temporal variability (Quincey et al., 2017). The aerodynamic roughness length is a key parameter for the calculation of turbulent fluxes (Chambers et al., 2020), since a change in $z_0$ by an order of magnitude can double the estimated turbulent fluxes (Brock et al., 2006; Munro, 1989). The uncertainty in $z_0$ values therefore presents a serious challenge for the calculation of surface ice melt (Smith et al., 2016) and its accurate parameterization, especially for complex ice surfaces, is crucial (Quincey et al., 2017).

The objective of this study is to assess the application of UAVs for capturing spatially variable $z_0$ values of heavily crevassed ice surfaces. A photogrammetry method is used to build DEMs of the mapped glaciers from the aerial images, which are then utilized to estimate the aerodynamic roughness length of crevassed tidewater glaciers in Svalbard. The main advantage of this



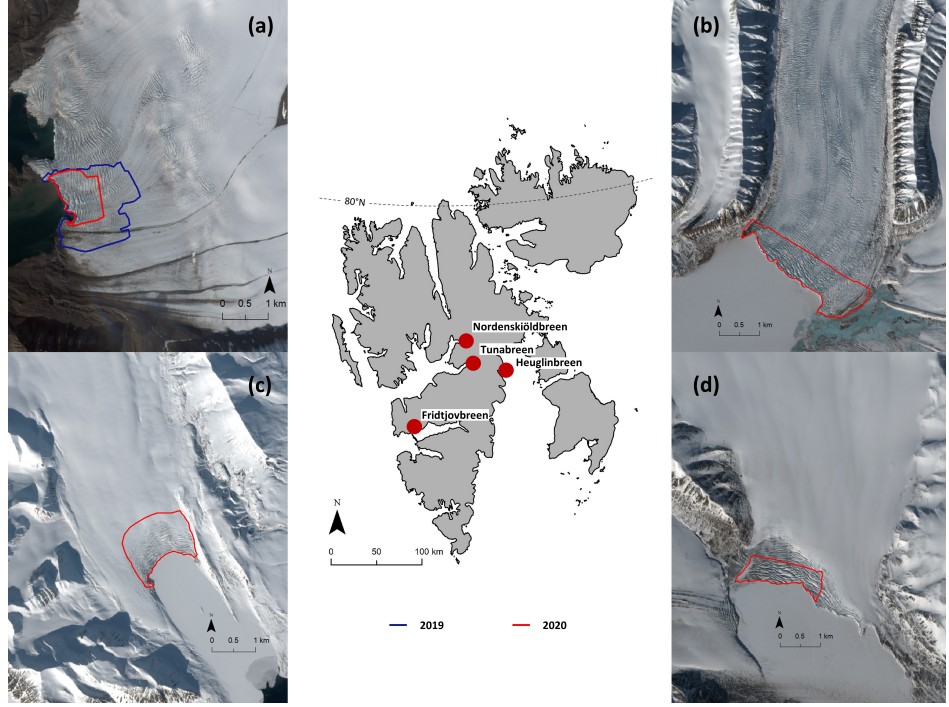

**Figure 1.** Field sites: Map of Svalbard with marked locations of the investigated tidewater glaciers (red dots). Additionally, Sentinel-2 satellite images (ESA, 2020) of the glaciers Nordenskiöldbreen (a), Tunabreen (b), Fridtjovbreen (c) and Heuglinbreen (d) taken at the according week of fieldwork provide a closer look. The lines mark the mapped front area for each glacier in 2019 (blue) and 2020 (red).

approach is that increasingly available UAV technology is used to estimate turbulent fluxes that are usually difficult to measure in the field. Furthermore, the chosen DEM approach allows glaciers to be divided into areas of different aerodynamic roughness length values, leading to a better spatial representation of the turbulent fluxes and therefore surface ice melt on glaciers.

## 2 Data and methods

The following section describes how a DEM was generated from aerial imagery of crevassed glaciers in Svalbard. Images
were obtained using off-the-shelf UAVs. In addition, different methods to calculate the aerodynamic roughness length from the DEMs are introduced.

### 2.1 Field sites

Four heavily crevassed tidewater glaciers in Central Spitzbergen (see Fig. 1 and Table 2) were visited during three field campaigns. Nordenskiöldbreen was visited in summer 2019 and 2020. In spring 2020 further fieldwork was conducted on
Fridtjovbreen, Heuglinbreen and Tunabreen. Fridtjovbreen is a single tidewater glacier of about 13 km length, flowing south-



**Table 2.** Overview of the tidewater glaciers visited and the date of their UAV survey. Additionally, the size of the mapped area and DEM resolution, the average height and length of roughness elements for a up-/down-glacier (cross-glacier) wind direction are listed.

| Glacier | Date | Location | Area (km$^2$) | Res. (m/px) | Height (m) | Length (m) |
|---|---|---|---|---|---|---|
| Nordenskiöldbreen 2019 | 19.-22. August 2019 | 78°39' N, 17°00' E | 2.6 | 0.17 | 10 (7) | 29 (41) |
| Tunabreen | 28. April 2020 | 78°27' N, 17°23' E | 2.4 | 0.19 | 14 (11) | 37 (43) |
| Heuglinbreen | 04. May 2020 | 78°21' N, 18°47' E | 1.1 | 0.21 | 14 (9) | 31 (35) |
| Fridtjovbreen | 07. May 2020 | 77°47' N, 14°31' E | 1.0 | 0.28 | 8 (6) | 29 (46) |
| Nordenskiöldbreen 2020 | 21. July 2020 | 78°39' N, 17°00' E | 0.9 | 0.22 | 16 (9) | 35 (49) |

wards and terminating in Van Mijenfjorden on the western side of Spitzbergen island. Here precipitation and temperature are relatively high with an annual temperature of about -4°C and precipitation of up to 1000 mm (Hanssen-Bauer et al., 2019). Both Nordenskiöldbreen and Tunabreen are outlet glaciers (flowing from northeast to southwest) draining the large Lomonosovfonna ice cap, where the precipitation usually is lower than on the west coast (Hagen et al., 1993). While Norden-

skiöldbreen is a roughly 15 km long, wide tidewater glacier terminating in Billefjorden, Tunabreen is narrower, with a length of about 20 km and terminating in Tempelfjorden. Additionally, both Tunabreen and Fridtjovbreen are known to have experienced a surging event. While the surge on Fridtjovbreen already happened during the 1990s (Murray et al., 2012), Tunabreen surged more recently in the years 2003 to 2005 and had another advance of the glacier front about ten years later (Ericson et al., 2019). Heuglinbreen is a tidewater glacier flowing southwards and terminating into Mohnbukta, a bay on the east coast of Spitzbergen.

This region is known to be particularly cold (annual temperature of about -10°C) and rather dry (annual precipitation up to 700 mm) (Hanssen-Bauer et al., 2019).

## 2.2    Field data collection

UAV-based aerial imagery was collected during each fieldwork campaign with off-the-shelf UAVs (a DJI Mavic 2 Enterprise and a DJI Phantom 4 Pro). In order to have a sufficient resolution of the crevasse fields, a target DEM-resolution of about 0.25

m/px was chosen. To achieve this DEM resolution, the UAV imagery aimed for a ground sampling distance (GSD) of at least 0.1 m/px, a forward overlap of 90 %, and a side overlap of 80 %. During fieldwork the UAVs were planned to operate at an altitude of 200 m above ground-level, taking nadir-viewing pictures. Flights were conducted with pre-programmed waypoints and a run separation of 70 m. Since information about the glacier surface elevation was unknown to the UAV, the true altitude above the glacier surface was typically less than 200 m. As a result, the GSD ranged between 0.04 - 0.07 m/px and the subsequent

DEM resolution ranged between 0.17 to 0.28 m/px (see Table 2).

## 2.3    DEM generation and preparation

The high-resolution images from the UAVs were processed with a structure-from-motion (SfM) multi-view stereo (MVS) photogrammetry method using Agisoft Metashape version 1.6.2 (Agisoft LLC, 2020). Building DEMs in Agisoft is a three-





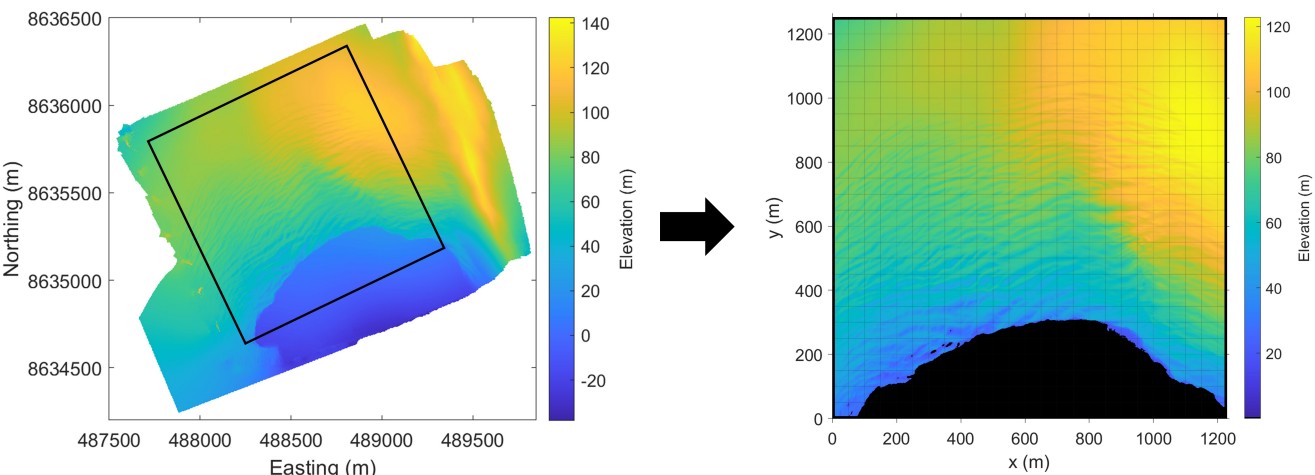

**Figure 2.** The originally mapped DEM (left) of Fridtjovbreen was rotated and cropped (black frame) before using it for the model calculations. Additionally, the elevation values of the sea ice area were removed and the whole DEM was divided into sub-grids of 50 m × 50 m (right).

step process: image alignment, construction of a dense cloud, and DEM generation (Verhoeven, 2011). The software runs on a
fully automatically workflow. However, the whole processing comes along with many parameter settings that can be selected to
improve the DEM quality depending on the input data and the output purpose. To determine an ideal set of parameters for our
approach, many different combinations of settings were tested leading to the optimized final settings (see Dachauer (2020)).

Before starting the model calculations of the aerodynamic roughness length, the DEMs obtained from the SfM processing
were rotated in such a way that the glacier flow direction corresponds to the column alignment of the DEM, with the front at
the lower part (see Fig. 2). This allowed the estimation of $z_0$ values for all four dominant wind directions (downwind, upwind,
crosswinds). Furthermore, the DEMs were cropped to a shape that contains the most crevassed zones. This area is specific
to each glacier and varies significantly due to the differences in glacier size, time available for mapping, number of UAV
batteries, and limitations due to GPS interference. The sea ice and water area in front of the glaciers was removed, since it has
no contribution to the turbulent heat exchange of the glacier.

This study assumes that the mean airflow is blowing parallel to the slope of the glacier. This means that the glacier slope
has no effect on the aerodynamic roughness (Fitzpatrick et al., 2019). Furthermore, it means that the aerodynamic roughness
is in the first place influenced by the macro-structure of the surface (crevasses, large obstacles, etc.) and not by small-scale
surface roughness on the crevasse obstacles. Accordingly, the selected grid size must be selected in a way that represents this
macro-structure of the surface, since small-scale roughness elements do not represent the real topographic expression. Linear
detrending over long baselines manages to represent areas of high curvature (Smith, 2014) and is therefore appropriate for





this purpose. Figure 3 presents the impact and importance of the chosen sub-grid size (i.e. length of detrended transect) on the modelled surface roughness. The blue line on Fig. 3 (a) shows a random transect of two roughness elements of 30 m width on Fridtjovbreen. Two linear detrending methods were applied to the surface data of this illustration. First, the green line detrended the whole transect. Second, the transect was detrended in 10 m intervals (purple line). The detrended data (Fig. 3 (b)) shows that the 10 m grid size only captured the small-scale surface roughness of the obstacles (purple line). In contrast, the linearly detrended transect length of 60 m (green line) managed to represent the two large roughness elements. Therefore, a sub-grid size of at least the width of an average obstacle should be chosen to account for the macro-structure surface roughness of the crevassed glaciers. On average, the mapped tidewater glaciers had obstacle widths between 30 m and 50 m (calculated according to the transect method of Munro (1989), see Table 2). Thus, the final DEMs were subdivided into rectangular sub-grids of 50 m × 50 m (grid in Fig. 2), each estimating one $z_0$ value.

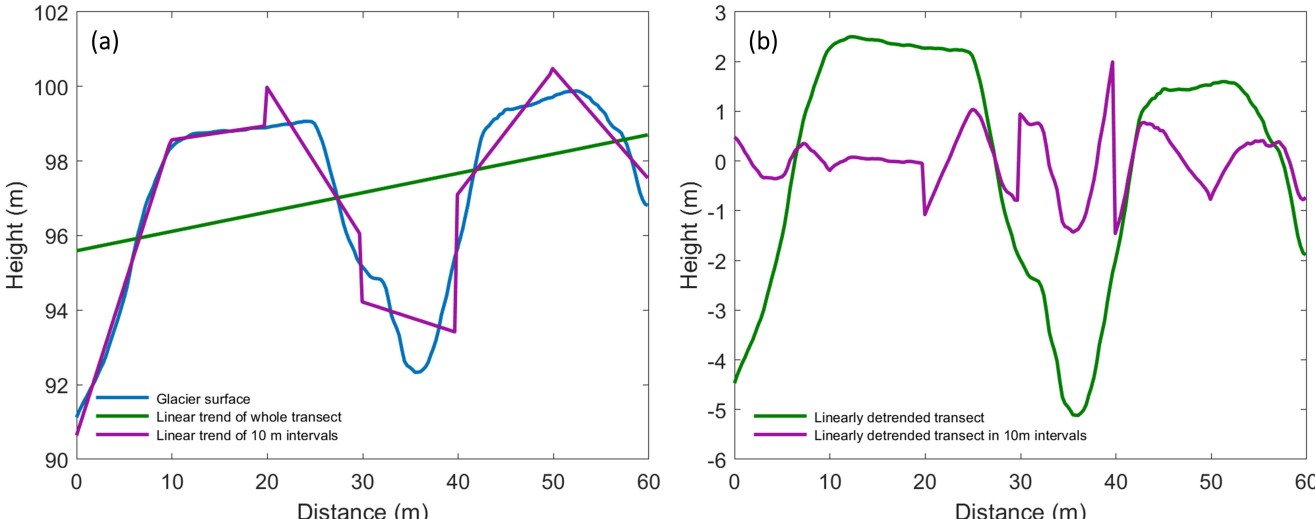

**Figure 3.** Graph (a) shows a random transect of two roughness elements on Fridtjovbreen (blue line). The transect is linearly detrended, either over the whole profile (green) or in 10 m intervals (purple). Graph (b) illustrates the linearly detrended surface data applied at the whole transect (green line), or applied at 10 m intervals (purple line).

## 2.4 Models for aerodynamic roughness length estimation

The aerodynamic roughness length was calculated for each sub-grid DEM and all wind directions with five models after 2D-linear detrending (see Fig. 3). The most common models in glaciology for calculating $z_0$ using the bulk method are based on the work of Lettau (1969), who developed the following equation for the bulk aerodynamic roughness length:

$$z_0 = c_d h^* \frac{s}{S}, \tag{1}$$





where $h^*$ is the effective obstacle height (m), $s$ is the silhouette or frontal area of the obstacle (m$^2$) and $S$ the horizontal ground area (m$^2$). The value $c_d = 0.5$, first proposed by Lettau (1969), corresponds to the average drag coefficient of a characteristic roughness element.

The definition of the parameters of Equation 1 turns out to be more difficult in glacial environments where the individual roughness elements vary a lot in height, size and density (Chambers et al., 2020). Therefore, the original equation has been adjusted and further developed in several studies, leading to the five different models used in this study (Table 3). The five models can be subdivided into two groups according to how they determine and measure roughness elements. One group counts the number of roughness elements in a transect (hereafter called the 'transect method'), while the other group is based on a raster approach (hereafter called the 'raster method') using every DEM cell value for the calculation of $z_0$.

Two models using the transect method were included in this study, only differing in their definition of the effective obstacle height $h^*$ (m). Each row of the detrended sub-grid was treated as a separate transect of length $X$ (m), whereof the transition frequency $f$ from below to above the mean elevation was recorded. First, the Lettau model calculates $h^*$ by taking the average vertical extent of the detrended roughness elements, as described by Lettau (1969). Second, the Munro model simplifies Equation 1 of Lettau (1969) such that the height of roughness element $h^*$ is calculated by taking twice the standard deviation of elevations along the detrended transect, as described by Munro (1989). In contrast to the study of Munro (1989), which used wind-perpendicular surface transects for the roughness calculation, we used wind-parallel profiles for both models of the transect method. This adaptation is essential for heterogeneous roughness elements like crevasses which are naturally streamlined (Smith et al., 2016) since wind systems are influenced by the large-scale catchment topography and therefore often flow up or down the glacier (Quincey et al., 2017). The transect method presents a simple approximation of the roughness elements across a profile and, in contrast to the raster method, assumes all roughness elements to be of equal height, uniformly distributed, isotropic and not affected by any sheltering effects (Smith et al., 2016).

The raster method is directly based on the Lettau (1969) Equation 1 and the elevation differences between two adjacent cells define the surface roughness at the end. Three models following the raster method were included in this study. First, the Smith model was based on the 'DEM-based' approach described by Smith et al. (2016). The effective obstacle height $h^*$ was calculated as the mean elevation of all the cells of each row above the zero plane. The second model of the raster method (hereafter called Chambers model) was based on the 'DEM method' described by Chambers et al. (2020). Its only difference to the Smith model is again the definition of the effective obstacle height $h^*$, which is twice the standard deviation of elevations above the detrended plane. The third model of the raster method (hereafter called Fitzpatrick model) was based on the 'Block estimation' described by Fitzpatrick et al. (2019). While the two previous raster methods calculated some of the parameters row-wise, this method followed a moving-window approach. The obstacle height $h^*$ corresponded to the mean of all the detrended elevation values above the zero plane within the window. In this study, a window size of 30 m was chosen, which corresponded to the lower estimation of an average roughness element on the investigated glaciers (see Table 2). For all three models, $s$ and $h^*$ were calculated individually for each row of sub-grid (moving-window for the Fitzpatrick model, respectively). Accordingly, the ground area $S$ was assigned to the area of the sub-grid (moving-window) and the final $z_0$ value for each sub-grid then calculated by taking the mean of its row (moving-window) values.





**Table 3.** Overview of parameters from the Lettau (1969) Equation 1 used for the $z_0$ calculation of each model.

| Parameter | Smith | Chambers | Fitzpatrick | Munro | Lettau |
|---|---|---|---|---|---|
| Effective obstacle height $h^*$ (m) | Mean height above the detrended plane | $2 \times$ standard deviation of $z$ above detrended plane | Mean height above the detrended plane | $2 \times$ standard deviation of detrended profile | Mean obstacle height |
| Silhouette area $s$ (m$^2$) | Frontal area above detrended plane across whole sub-grid calculated for each cardinal wind direction | | Frontal area across window above the height of the first row cells calculated for each cardinal wind direction | Frontal area of a modelled roughness element: $h^* X/2f$. With $X$ = transect length, $f$ = transition frequency | |
| Ground area $S$ (m$^2$) | Full area of DEM sub-grid | | Area of moving-window | Ground area of a modelled roughness element: $(X/f)^2$ | |
| Drag coefficient $c_d$ | 0.5 | | | | |
| Roughness length $z_0$ (m) | $z_0 = c_d h^* \frac{s}{S}$ | | | | |

## 3 Results

The DEMs obtained from the UAV-based imagery and processed with the SfM-MVS method illustrate that the crevasses of the mapped glaciers are in general aligned perpendicular to the glacier flow direction. The crevasses closer to the front are often deeper and larger in terms of spacing (width of crevasse opening) compared to crevasses located upstream the glacier

155 (see Fig. 4). The DEMs were then used to calculate the aerodynamic roughness lengths with the two transect method models and the three raster method models. The results (e.g. Fig. 4, 6 and 7) show that the spatial variability of $z_0$ values across the mapped area is up to three (locally even four) orders of magnitude. In general, the larger a roughness element the larger its aerodynamic roughness length. The highest values were calculated with the transect methods and for winds blowing parallel to the flow direction of the glacier. Such values can be found close to the glacier front where crevasses are big and steep. The

160 lowest values are estimated with the raster methods for smooth, crevasse-free ice and for cross-glacier wind directions.

### 3.1 Model results

Figure 4 shows the estimated aerodynamic roughness length values for each of the five applied models for the down-glacier wind direction on Nordenskiöldbreen 2019. The results reveal that all models agree on the relative spatial $z_0$ patterns across the glacier. Accordingly, for the flatter and less crevassed part (e.g. to the right of the mapped area on Nordenskiöldbreen)

165 all models show lower sub-grid $z_0$ values (red) compared to the heavily crevassed part close to the glacier front (yellow). To




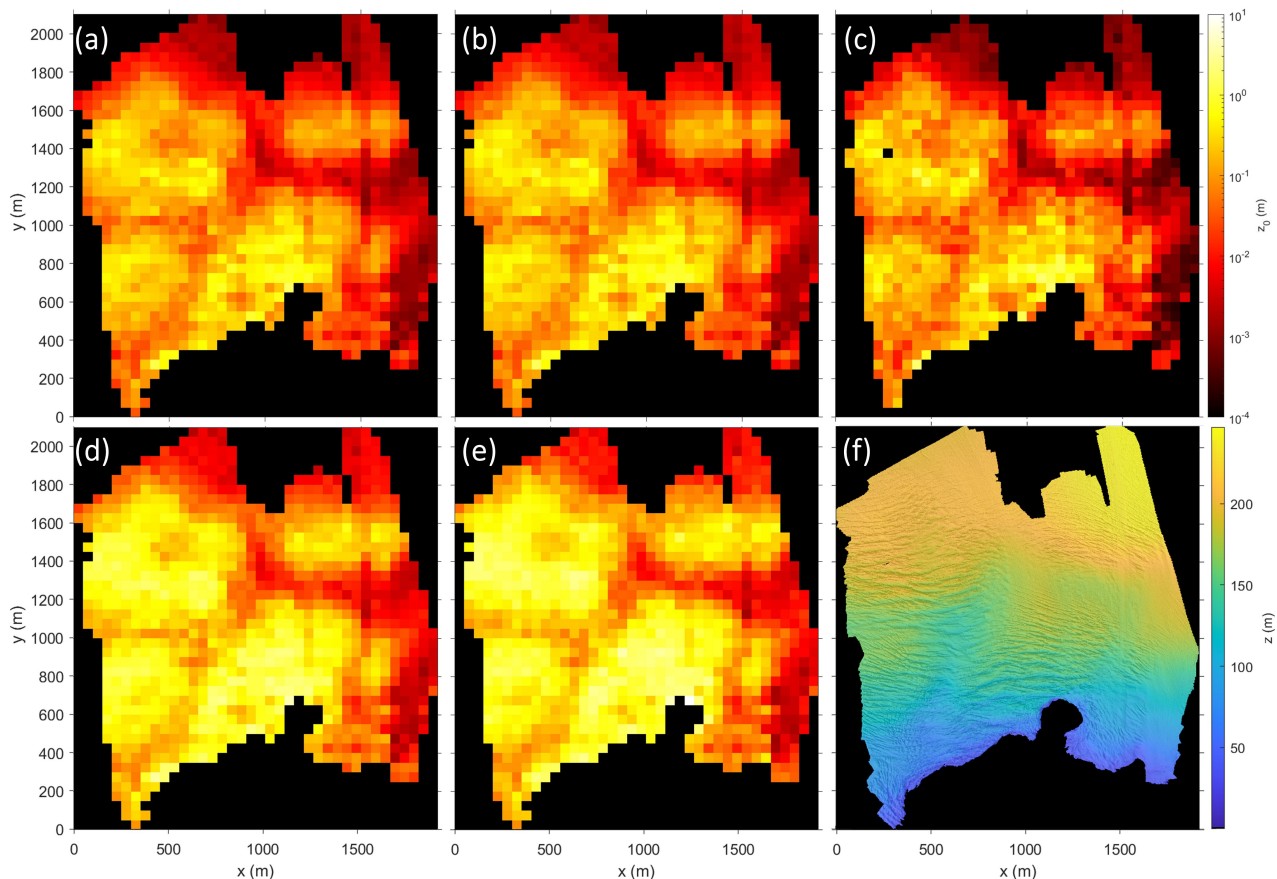

**Figure 4.** Variability of $z_0$ values for Nordenskiöldbreen (2019) depending on the calculation models Smith (a), Chambers (b), Fitzpatrick (c), Munro (d) and Lettau (e) for a sub-grid size of 50 m × 50 m and a down-glacier wind direction. Graph (f) shows the DEM with the underlaid hillshade layer.

further investigate the relative agreement between the models a statistical correlation test on the $z_0$ data of Nordenskiöldbreen (averaged in all wind directions) was conducted. The correlation test showed that all models are strongly correlated to each other leading to $R^2$ values of 0.877 and higher (Dachauer, 2020). A similar correlation has also been observed on the other glaciers.

170    Figure 4 further illustrates that the absolute values of the models Munro and Lettau show higher roughness values on the same sub-grid than the other three models. In more detail, the Lettau model estimates are generally higher than those of the Munro model and Chamber $z_0$ values higher than those of the Smith model. The three models of the raster method provide sub-grid $z_0$ values of about 0.001 m for slightly crevassed areas and 1 m for really heavily crevassed areas. The same sub-grids calculated with the two transect methods produced $z_0$ values which are locally up to one order of magnitude larger. Table 4 175    illustrates the down-glacier and cross-glacier (left-to-right) mean $z_0$ values for all glaciers and models. Table 4 also shows that





**Table 4.** Overview of mean $z_0$ values (m) for each glacier and model either for the down-glacier or cross-glacier (left-to-right) wind direction.

| Glacier | Wind Direction | Smith | Chambers | Fitzpatrick | Munro | Lettau |
|---|---|---|---|---|---|---|
| Nordenskiöldbreen 2019 | Down-glacier (m) | 0.129 | 0.151 | 0.126 | 0.393 | 0.496 |
| | Cross-Glacier (m) | 0.039 | 0.046 | 0.036 | 0.124 | 0.155 |
| Tunabreen | Down-glacier (m) | 0.223 | 0.271 | 0.196 | 0.709 | 0.883 |
| | Cross-Glacier (m) | 0.092 | 0.113 | 0.090 | 0.305 | 0.363 |
| Heuglinbreen | Down-glacier (m) | 0.150 | 0.187 | 0.175 | 0.415 | 0.424 |
| | Cross-Glacier (m) | 0.045 | 0.054 | 0.051 | 0.141 | 0.156 |
| Fridtjovbreen | Down-glacier (m) | 0.082 | 0.098 | 0.115 | 0.248 | 0.276 |
| | Cross-Glacier (m) | 0.028 | 0.034 | 0.039 | 0.090 | 0.094 |
| Nordenskiöldbreen 2020 | Down-glacier (m) | 0.231 | 0.276 | 0.213 | 0.706 | 0.867 |
| | Cross-Glacier (m) | 0.052 | 0.062 | 0.050 | 0.172 | 0.212 |

average $z_0$ values across the glacier vary almost up to half an order of magnitude between the models. In summary, despite the clear relative agreement between the models, the estimated magnitude of the $z_0$ values varies substantially between the models - especially between the raster and transect method models.

## 3.2 Wind direction variability

Figure 5 illustrates the map of Fridtjovbreen with $z_0$ values obtained with the Smith method for all four cardinal wind directions. The results show that $z_0$ values are higher for wind directions that face the crevasses perpendicularly (i.e. up- and down-glacier) and lower for wind that blow parallel to the elongated crevasse features (i.e. cross-glacier). Since crevasses are mainly oriented perpendicular to the glacier flow direction, the mapped areas show a strongly anisotropic pattern of the glacier surface. This wind dependency effect is visible on all five applied models and independent of the roughness element size. However, Fig. 5 illustrates that larger roughness elements (i.e. area close to the glacier front and crevassed area in the centre of Fridtjovbreen) present a stronger wind dependency because they vary more strongly with changing wind directions compared to areas that are less crevassed (e.g. upper part of Fridtjovbreen).

Additionally, Fig. 6 shows the boxplot graph of $z_0$ estimations illustrating the wind direction dependency of $z_0$ values on Fridtjovbreen. The boxplot $z_0$ medians of all models vary in a range of 0.012 m to 0.037 m for crosswind directions and 0.058 m to 0.2 m for up- and down-glacier wind systems. The mean and median values between up- and down-glacier wind systems (left-to-right and right-to-left crosswinds, respectively) never differ more than 10 % and only rarely more than 5 % independent of the chosen model or glacier. In summary, the wind direction has a large impact on the resulting aerodynamic roughness length values. Its effect on average or median $z_0$ values slightly exceeds the model variability. Locally however, both parameters can vary about one order of magnitude.



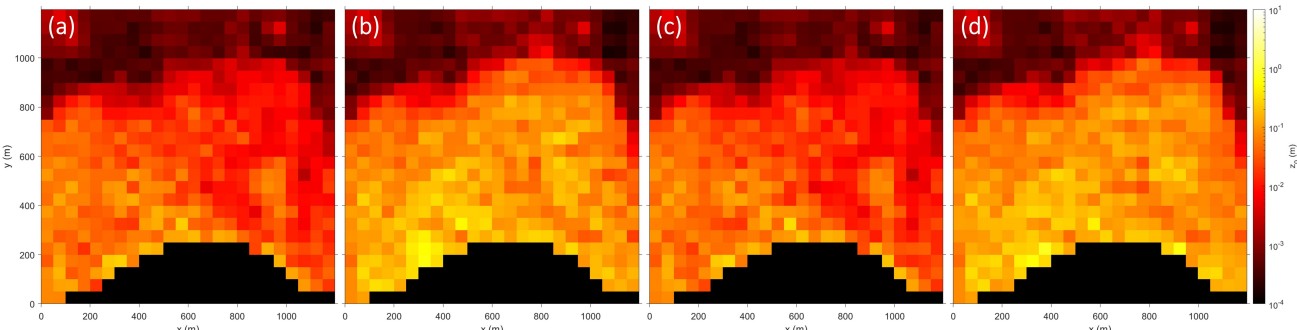

**Figure 5.** Variability of $z_0$ values for Fridtjovbreen depending on the wind direction for a sub-grid size of 50 m calculated with the Smith model. Winds blowing across the glacier either from the left-to-right (a) or right-to-left (c) produce smaller $z_0$ values than down- (b) or up-glacier (d) wind systems.

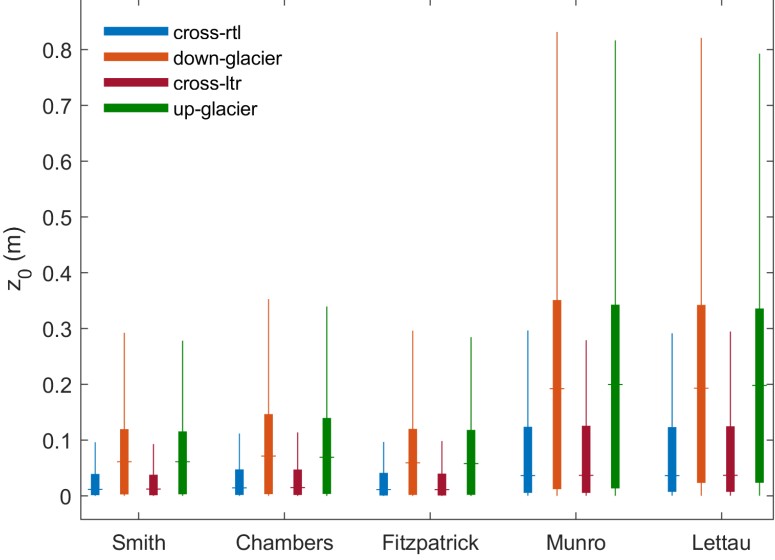

**Figure 6.** Boxplot visualization of sub-grid $z_0$ values for all four wind directions and each applied model determined on Fridtjovbreen. The wind direction is either down- (orange) and up-glacier (green) or cross-glacier from right-to-left (blue) and left-to-right (red).

### 195  3.3  Glacier variability

All glaciers showed a similar range of $z_0$ values with decimeter to meter scale for heavily crevassed areas and milimeter to centimeter scale for less crevassed areas (see Fig. 4, 5 and 7). However, the results of Table 4 show that the mean $z_0$ values for the Smith model and down-glacier wind direction are somewhat larger on Tunabreen (0.223 m) and the extract of





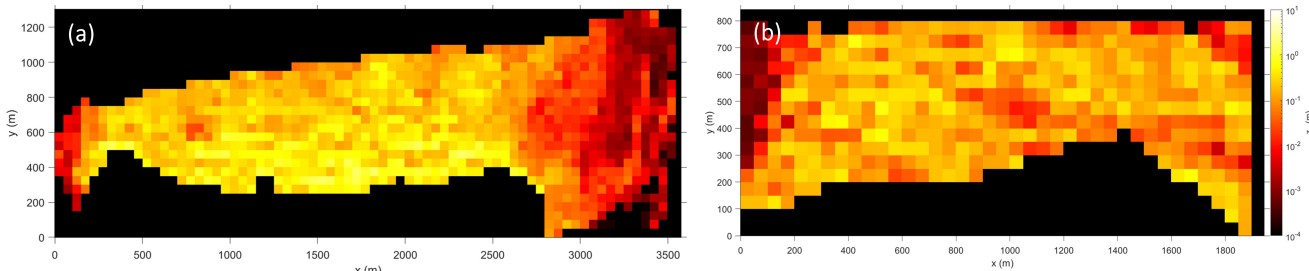

**Figure 7.** Variability of $z_0$ values for the glaciers Tunabreen (a) and Heuglinbreen (b) calculated with the Smith model for a down-glacier wind direction and a sub-grid size of 50 m.

Nordenskiöldbreen mapped in 2020 (0.231 m) compared to the other glaciers. The same mean $z_0$ values on Nordenskiöldbreen

measured in 2019 are lower with only 0.129 m. Heuglinbreen (0.15 m) and especially Fridtjovbreen (0.082 m) show lower mean $z_0$ values than the glaciers mentioned above. Nevertheless, all the observed patterns recognized in the investigations of this study (e.g. impact of wind direction or sub-grid size on $z_0$ values) are independent of the chosen glacier and vary (if even) only in magnitude rather than relative patterns in between the glaciers.

On a side note, it is interesting to study the results of Nordenskiöldbreen 2019 and 2020 in more detail. The comparison

provides insights on the inter-annual temporal variability of $z_0$ values for two consecutive years. The mapped area in summer 2020 (one day of fieldwork) was smaller compared to the field campaign approximately one year earlier in 2019 (three days of fieldwork). Therefore, only the overlapping area has been used for the temporal variability investigation and the DEM extract of Nordenskiöldbreen 2019 (0.17 m/px) was resampled to the resolution of the Nordenskiöldbreen 2020 DEM (0.22 m/px). $z_0$ values on Nordenskiöldbreen are very similar for the two consecutive years. The mean $z_0$ value for the Smith model and

down-glacier wind direction in 2019 is with 0.25 m only slightly higher than the corresponding value of 0.23 m on the same area but one year later. The observations are in line with other studies (i.e. Fitzpatrick et al., 2019), which also did not observe a large difference in $z_0$ estimates for the same location measured in two consecutive years. Relatively, the mean $z_0$ values never deviated more than 20 % and mostly less than 10 % between the two years. This small deviations in $z_0$ values makes it hard to find the real reason for the decrease in $z_0$ estimations. Nevertheless, the differences might be explained due to the fact that

the data for the DEM 2020 was collected one month earlier. Thus, there might still be some snow bridges remaining in the crevasses what could have lowered the resulting aerodynamic roughness length. In summary, the differences in produced $z_0$ estimations across the used models exceed the temporal $z_0$ variability by far, independent of the model calculation.

### 3.4 Sub-grid size dependency

The mapped glacier areas were divided into sub-grids with a grid size of 50 m × 50 m to account for the spatial variability

of $z_0$ on the glacier. However, to investigate the grid size dependency of the sub-grids on $z_0$ values, a small case study on the glacier Fridtjovbreen was conducted. The Smith model with a down-glacier wind direction was used to calculate aerodynamic



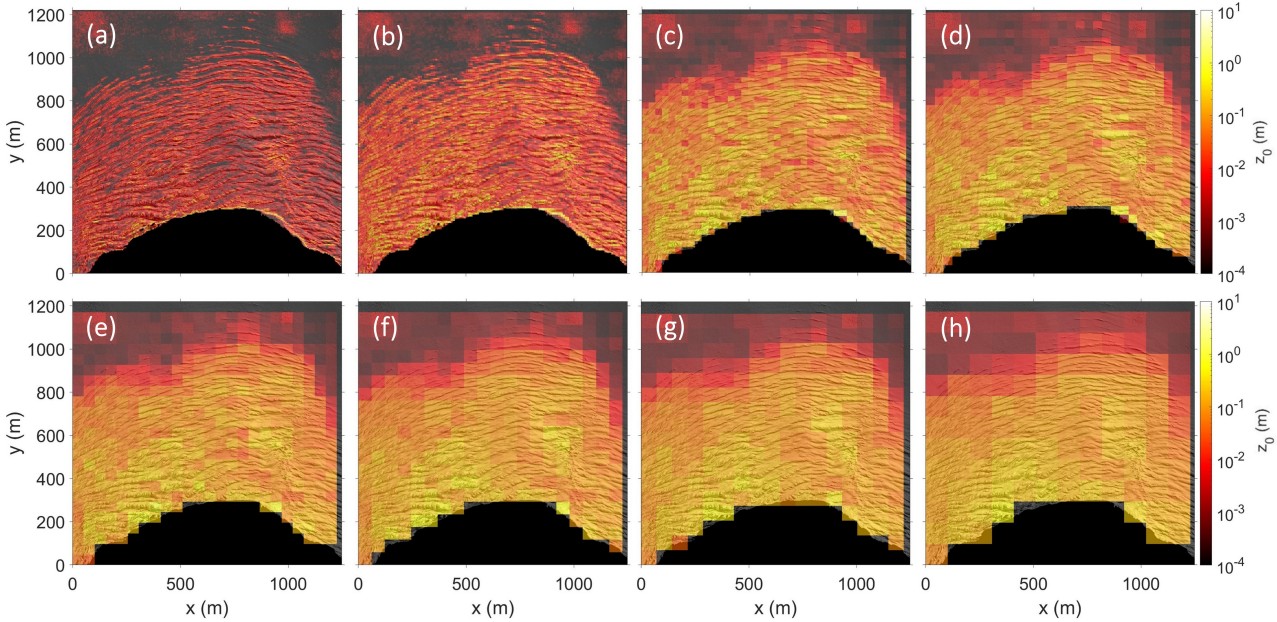

**Figure 8.** Scale dependent $z_0$ values for the Smith model and down-glacier wind direction applied on Fridtjovbreen for sub-grid size resolutions of 5 m (a), 10 m (b), 30 m (c), 40 m (d), 50 m (e), 60 m (f), 70 m (g) and 100 m (h) with underlaid hillshade layer for orientation.

roughness lengths for sub-grid sizes between 5 m and 150 m. Figure 8 illustrates the scale dependency of $z_0$ values to the selected sub-grid size. The results show that a grid size of 5 to 10 m mostly produces $z_0$ values in a scale of centimeters. Between sub-grid sizes of 10 and 50 m a higher grid size results in higher $z_0$ values. From 50 m onwards, the $z_0$ values are

mostly in the scale of decimeter and do not change substantially. This behavior was also observed with the other models and for all wind directions. For small sub-grid sizes, the $z_0$ values are likely representing microtopography rather than the macro-scale surface roughness of the crevasses.

Figure 9 summarizes the mean $z_0$ results of the mentioned case study for the down-glacier wind direction. For all the models, the mean $z_0$ values increase by at least half order of magnitude between a sub-grid size of 5 m and 150 m. All models show

a similar pattern with strongly increasing $z_0$ values for small grid sizes (5 m to 30 m) and only slightly increasing estimates afterwards. Grid sizes of 70 m or more are loosing their strong scale dependency effect leading to stable $z_0$ estimations. The same effect is visible on Fig. 8, where the chosen grid size of 50 m represents similar $z_0$ values (colors) as the higher grid sizes for the same location while still providing a considerable grid resolution.

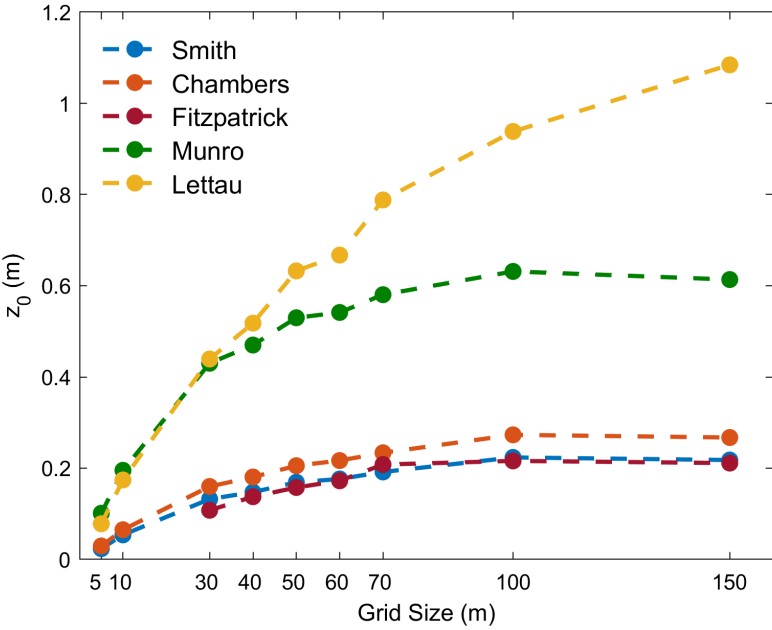

**Figure 9.** Scale dependency of mean $z_0$ values for the five applied models and a down-glacier wind direction on Fridtjovbreen with chosen sub-grid sizes from 5 m to 150 m. The $z_0$ values are increasing with larger grid size independent of the chosen model.

## 4 Discussion

### 4.1 Model inputs for aerodynamic roughness length estimation

#### 4.1.1 Validation of digital elevation model accuracy

The obtained DEM resolution with the SfM-MVS method (about 0.25 m) was accurate enough to capture the large crevasse structures. Given the advantages of an UAV compared to other devices (e.g. cheap price, applicable in inaccessible areas (Hann et al., 2021)), this study shows that UAVs provide a reliable and effective way of data gathering for aerodynamic roughness length estimation on glaciers. Nevertheless, the depth of the crevasses must be seen as a minimum depth and the crevasses might penetrate further into the glacier than actually measured. This is due to snow-bridges or the lack of reflected light from the deep crevasses. The latter prevents the SfM-MVS methods to correctly construct the deeper parts of the crevasse. Additionally, the lowest points of the crevasses are very narrow and may not be captured accurately (Ryan et al., 2015). However, in an aerodynamic context those narrow crevasses are not likely to have a significant influence on the heat exchange since they lie below the penetration depth of effective turbulent mixing (Nicholson et al., 2016).

In the scope of this study the use of ground control points (GCPs) was also considered. GCPs can significantly increase the georeferencing accuracy of the DEMs (Chudley et al., 2019). However, it was practically not possible to place GCPs on the crevassed glacier surfaces, due to safety reasons. Therefore, the georeferencing information was provided only from the



on-board GPS and the measurements of camera orientation (James et al., 2017). In other words, the positional accuracy of the
DEM is limited by the internal GPS system of the UAV, which has a relatively low accuracy (Federman et al., 2017). However,
since the objective of this study was not to obtain a high-accuracy DEM, but rather to investigate the effect of relative distances,
the given hover accuracy of $\pm$ 1.5 m horizontally and $\pm$ 0.5 m vertically for both UAVs (DJI, 2017, 2019), was considered
sufficient (Dachauer, 2020).

Nevertheless, a comparison with Sentinel-2 satellite data (ESA, 2020) revealed horizontal positioning errors (data not
shown). The deviation was classified as a systematic offset (for more details see Dachauer (2020)). The detected small horizon-
tal distortions of about 1.7 % mainly occurred on the ends of the DEMs. This is a typical feature appearing when only using
nadir imagery and is related to self-calibration, because the reconstruction software is not able to derive the accurate radial
lens distortion leading to a systematic 'doming' DEM deformation (James and Robson, 2014). However, a systematic error is
of low significance for this study since only a low relative accuracy of the roughness elements (i.e. distortion) influences the
estimation of the aerodynamic roughness lengths. Furthermore, the influence of a small distortion of a few percents or several
meters across the mapped area on the resulting $z_0$ values is minor compared to other parameters such as wind direction, model
calculation or scale dependency. In more detail, a distortion of 2 % led to a change in $z_0$ of about 4 % (Dachauer, 2020). This
means that the obtained DEMs in this study are a reliable data source to estimate the aerodynamic roughness length.

### 4.1.2 Scale dependency

Many studies investigated and encountered the dependency of $z_0$ estimations on the size of the sub-grid or the transect length
(e.g. Fitzpatrick et al., 2019; Miles et al., 2017; Rees and Arnold, 2006) and reported that larger sub-grids or longer transects
cause $z_0$ values to increase (Chambers et al., 2020). Also the case study conducted on Fridtjovbreen data revealed that the $z_0$
values increase with larger grid size independent of the chosen glacier, model or wind direction. This can be explained by the
fact that glaciers often have heterogeneous roughness elements (Quincey et al., 2017). In general, it needs to be considered
that the selection of an appropriate grid size comes with a large potential uncertainty. To find the most meaningful grid size
a comparison with independent methods like aerodynamic wind profiles is recommended (Chambers et al., 2020). Since this
option was not available in our study, the validity of the chosen grid size has been evaluated theoretically. The grid size in our
models is correlated to the ground area $S$, whose definition requires the grid size to be the size of one individual roughness
element (Lettau, 1969), which is 30 m to 50 m (see Table 2). According to the theory of Smith (2014), the definition of
'roughness' is related to the grid scale that separates the grain roughness (representing the texture of a roughness element) and
the form roughness (corresponding to the form drag of the roughness element itself). The grid size, at which the transition
from grain to form roughness occurs, provides a useful reference point and can be found on a kink in the trend line of a figure
plotting $z_0$ values against grid sizes (Shepard et al., 2001; Smith, 2014). In our study, this transition is again located somewhere
between a sub-grid size of 30 m to 50 m (see Figure 9). Therefore, in this study the grid size of 50 m was justified such that it
is large enough to include an average obstacle size and small enough to provide a considerable spatial resolution of $z_0$ values
across the glacier.





## 4.2 Model outputs of aerodynamic roughness length estimation

### 4.2.1 Wind direction dependency

The wind direction has a large impact on the magnitude of the $z_0$ values on glaciers since they obtain many anisotropic
roughness elements (Chambers et al., 2020). Our results confirmed this statement and further revealed that larger roughness
elements in general present a stronger wind dependency (see Fig. 5). The wind dependency effect was additionally investigated
with the calculation of the anisotropy ratio $\Omega$ (see Smith et al. (2006) for further explanation):

$$\Omega = \frac{z_{0\parallel} - z_{0\perp}}{z_{0\parallel} + z_{0\perp}}, \tag{2}$$

where subscripts $\parallel$ and $\perp$ denote parallel and perpendicular to the ice flow direction, respectively. Figure 10 illustrates a
histogram of the sub-grid's $\Omega$ values on Fridtjovbreen for the Smith (blue) and Munro (green) model. The results show that
the sub-grid $z_0$ values are strongly anisotropic all across the glacier whereas wind directions parallel to the flow direction
are dominant (since values are mostly positive). Both the raster (here Smith model) and the transect method (Munro) show a
similar pattern with the highest frequency at the range of 0.4 to 0.7 (Smith) and 0.4 to 0.8 (Munro). Although Tunabreen has
still mostly positive ratio values it is the glacier with the least anisotropic behaviour of all investigated glaciers. The importance
of wind direction can be observed in many studies (e.g. Fitzpatrick et al., 2019; Munro, 1989; Smith et al., 2016) and is
found to be the strongest on ablation zones where elongated features like meltwater channels and crevasses are frequent. Thus,
wind directions that face these features perpendicularly lead to higher $z_0$ values due to an increased form drag (Fitzpatrick
et al., 2019). However, winds are in general very likely to blow in direction of the mean slope angle or in a down-slope wind
direction (e.g. Fitzpatrick et al., 2019; Karner et al., 2013) due to katabatically forced down-slope winds, which are common
over the glacier terminus (Munro, 1989). Therefore, the effect of roughness anisotropy on the variation of $z_0$ values and hence
on turbulent heat exchange might not be that important in many cases, because not all four wind directions have the same
likelihood to occur (Fitzpatrick et al., 2019).


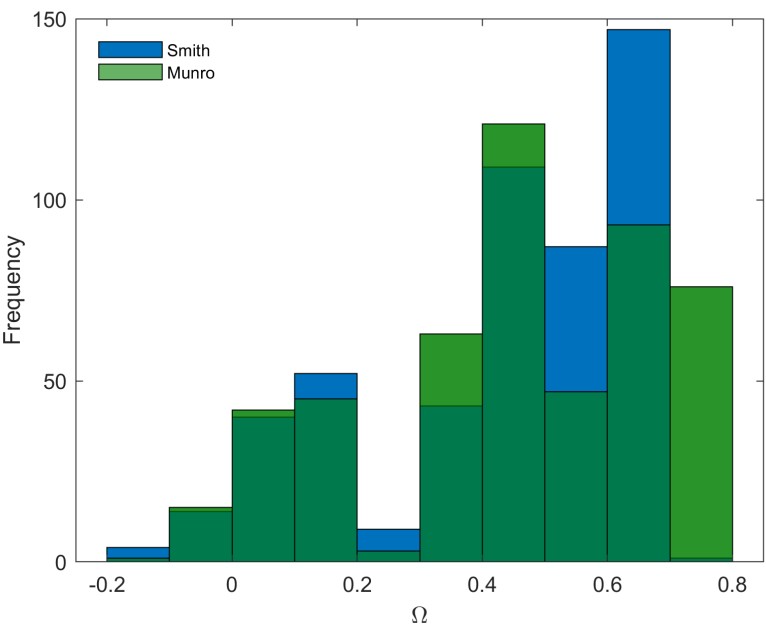

**Figure 10.** Anisotropy ratio values for the two models Smith (blue) and Munro (green) calculated for sub-grid $z_0$ values on Fridtjovbreen. A positive ratio towards 1 means that parallel winds (up- and down-glacier) are dominant over perpendicular winds (cross-glacier).

### 4.2.2 Variability across the glaciers

Since almost no values of $z_0$ for heavily crevassed glaciers are available in the literature, the validation of the results of this
study is difficult. The roughness elements investigated by previous studies (see Table 1) were smaller than the crevasses of this study. Therefore, it is expected that the $z_0$ values obtained here should be larger. This was the case, although there was a significant spread of $z_0$ values across the estimation models. The mean results in Table 4, and in particular their raster method results, fall mostly within the same order of magnitude. The range suggested by Fitzpatrick et al. (2019) fits most of the results from this study, across different methods and glaciers.

In general, Tunabreen and Nordenskiöldbreen (especially the part mapped in 2020) show higher $z_0$ values than Fridtjovbreen and Heuglinbreen. This is in good agreement with the average length and height of roughness elements estimated for each glacier in Table 2. The crevasses on Tunabreen and Nordenskiöldbreen are generally deeper and steeper than the crevasses on the other two glaciers. This might be explained by different dynamical behaviour, leading to higher $z_0$ values. In general, a faster flowing glacier leads to more crevasses in the terminus area of the glacier and a larger area of crevassing (Błaszczyk
et al., 2009). Additionally, Tunabreen was recently surging, what usually leads to very chaotically aligned crevasses (Mansell et al., 2012). Thus, these results are in line with the observation that the anisotropy ratio value $\Omega$ is lowest on Tunabreen, as perfectly aligned roughness elements increase the anisotropy effect. However, comparing average $z_0$ values among different





glaciers is challenging because the mean $z_0$ values depend a lot on the mapped input area and therefore the size of included roughness elements. The proportion of different glaciers that is heavily crevassed also varies markedly.

### 4.3 Model performance assessment

#### 4.3.1 Transect and raster method comparison

All models managed to detect the large spatial variability on glaciers and to produce high relative $z_0$ values in areas of large crevasses and low relative $z_0$ values for non-crevassed areas. The $R^2$ values among the models used in this study indicate that the $z_0$ values were correlated, but do not provide any conclusion about the quality of the individual models. Therefore, as discussed by Miles et al. (2017), each model manages to provide an approximation of roughness in relative scales. The $z_0$ results showed that the raster method produced values which are in general (locally up to one order of magnitude) lower than those calculated with the transect method. A series of studies (e.g. Chambers et al., 2020; Quincey et al., 2017; Smith et al., 2016, 2020) investigated the aerodynamic roughness length using both a transect and a raster method. However, no such study considered such heavily crevassed glaciers, which most likely makes the differences between the methods more obvious whilst also making it challenging to evaluate the model estimates. On rough or crevassed glacier ice, most of the mentioned studies also achieved lower $z_0$ values with the raster method compared to the transect method and the effect usually increased for larger roughness elements. This might be because the transect method does not account for sheltering of an obstacle (Smith et al., 2016). Furthermore, the raster method neglects the frontal areas below the detrended plane, assuming that it would be effectively sheltered. This plane indicates how far the effective turbulent mixing advances into the crevasses and the corresponding $z_0$ values are expected to be lower if the sheltering effect is considered (Nicholson et al., 2016).

#### 4.3.2 Sheltering effect

The sheltering issue can be discussed from another perspective. If the roughness elements on a plot are too densely packed, then the objects begin to interfere aerodynamically with each other (Rounce et al., 2015). They form a plateau-like new surface at their tops (Lettau, 1969) leading to a skimming flow. If this roughness density (frontal area divided by ground area) increases as far as is necessary to induce skimming flow, then the $z_0$ values decrease (Grimmond and Oke, 1999). Accordingly, the results from the transect methods which are not considering any sheltering effects are likely to overestimate the roughness, especially for densely packed obstacles (Nicholson et al., 2016). Several studies (e.g. Nicholson et al., 2016; Rounce et al., 2015; Smith et al., 2016) used a roughness density threshold of 20 % to 30 % (as stated by Macdonald et al. (1998)) for Equation 1 of Lettau (1969) to still be valid. In our study, all glaciers were tested with the Smith model for their roughness density. Apart from some single heavily crevassed sub-grids close to the glacier front which exceed the threshold of 30 %, all glaciers are below the given threshold with mean roughness density values between 0.1 and 0.15 for each glacier, indicating no skimming flow over the obstacles was likely. Therefore, this study shows that the Lettau (1969) Equation 1 can be assumed to hold also on heavily crevassed tidewater glaciers.





### 4.3.3 Obstacle height

The Smith and Chambers models, as well as Munro and Lettau models, only differ in the definition of $h^*$. Rounce et al. (2015) state that the lack of a clear obstacle definition presents the main problem of the bulk method approach introduced by Lettau (1969). Especially in crevassed glacier areas, where an apparent base level is missing, it is hard to define individual roughness elements and their according height (Nicholson et al., 2016; Smith et al., 2016). Moreover, Nield et al. (2013b) state that the obstacle height is the most important control parameter over the output of $z_0$, meaning that an appropriate definition of $h^*$ is

crucial. Smith et al. (2016) on the one hand question the rationale behind the standard deviation approach. The authors state that any chosen obstacle height will be somewhat arbitrary and that their definition (i.e. mean height above detrended plane) could be most meaningful on irregular ice surfaces. Chambers et al. (2020) on the other hand chose the standard deviation definition to preserve the influence of the larger roughness elements. To find out which model and which definition of $h^*$ might perform best, a comparison with alternative measurement methods (e.g. wind profiles) would be necessary, as done in several

studies (Fitzpatrick et al., 2019; Quincey et al., 2017; Smeets et al., 1999). For this study however, the area of interest was inaccessible and so this validation option was impossible. In any case, it clearly can be seen that the definition of $h^*$ has a lower impact on the results than the basic method itself (i.e. raster/transect).

### 4.3.4 Overall model performance

In summary, the transect method is assuming isotropy, equal size and distribution of roughness elements and not considering

sheltering. None of those assumptions is particularly valid on heavily crevassed tidewater glaciers. Additionally, the transect method assumes tower-like roughness elements whereas in reality crevasses are getting narrower with depth. The raster method is not making any of those assumptions. Furthermore, the raster method is making the best use out of the available dataset as every pixel value is considered for calculation (Smith et al., 2020). Despite the lack of any appropriate reference measurements, which clearly prevents a conclusion being drawn concerning best model performance, it can be stated that raster methods should

- at least theoretically - be more accurate. With this being the case, we have demonstrated that UAVs are the ideal platform for applying this method.

## 5   Conclusions

The heavily crevassed terminus areas of the tidewater glaciers Fridtjovbreen, Heuglinbreen, Nordenskiöldbreen and Tunabreen were mapped with UAVs to build DEMs that revealed crevasse shape information. To take into account the spatial variability

across the glacier, the DEM was divided into sub-grids of 50 m × 50 m, which was assumed to be large enough to include an average roughness element while still being small enough to account for the roughness variability across the glacier. Five different models (belonging to either the transect or the raster method) were applied to each DEM sub-grid to calculate the aerodynamic roughness length. The $z_0$ estimations from the transect method are in general greater (up to one order of magnitude) than the raster method estimations. Wind direction and sub-grid size have a large impact on the $z_0$ estimations, again
producing a range of up to one order of magnitude for each parameter. Winds blowing parallel to the ice flow direction produce larger $z_0$ values than cross-glacier winds. The chosen sub-grid size presents a large uncertainty in aerodynamic roughness length estimation. The resulting $z_0$ values are strongly scale dependent, such that a larger grid size leads to greater $z_0$ values. If all parameters (i.e. model, wind direction, grid size) are included, the spread of the resulting $z_0$ estimations is large, ranging from below one milimeter (snow-covered, smooth glacier surface) up to several decimeters (heavily crevassed ice) or locally

even more. Averaged $z_0$ values for down-glacier wind directions varied from 0.08 m to 0.88 m depending on glacier and model. Nevertheless, all models managed to detect the same spatial variability across the glacier. The UAV approach allows several $z_0$ values for each mapped glacier area to be derived, which is crucial for heavily crevassed glaciers, since one value would be a poor representation of the real roughness across such a complex topography. Therefore, models can now incorporate distributed $z_0$ estimations easily following UAV deployment, leading to a better representation of turbulent heat fluxes and prediction of

surface ice melt rates.

Spatial and temporal variability in crevassing, and a dependence on wind direction were found to extend the range of $z_0$ values found across tidewater glaciers. Variability caused by sub-grid size and model calculation assumptions reveal uncertainties which should be addressed by future investigations. Some degree of uncertainty also comes with the unsatisfactory georeferencing of the DEM in crevassed areas, because the inaccessible topography sets practical limitations, especially to

the use of GCPs. Furthermore, future work should seek a scale-independent method for $z_0$ calculation and also assess model performance using meteorological measurements (e.g. wind profiles) or computational fluid dynamics simulations.

*Code availability.* On demand

*Data availability.* On demand

*Author contributions.* Conceptualization, AJH, RH and AD; data curation, RH; methodology, AD and RH; project administration, RH;
formal analysis, AD; investigation, AD; visualization, AD; resources, RH and AJH; writing—original draft preparation, AD; writing—review and editing, AD, RH, AJH; supervision, RH and AJH. All authors have read and agreed to the published version of the manuscript.

*Competing interests.* The authors declare that they have no conflict of interest.

*Acknowledgements.* Fieldwork was funded by the Arctic Field Grant (RiS ID: 11148), provided by the Svalbard Science Forum (SSF) which is coordinated by the Research Council of Norway (RCN). The work is partly sponsored by the RCN through the Centre of Excellence
funding scheme, project number 223254, AMOS, and project CLIMAGAS with project number 294764.



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
