# Peer review of "Aerodynamic roughness length of crevassed tidewater glaciers from UAV mapping"

_The Cryosphere, 2021_

## Referee Comment (RC1)

**Review of the paper: Aerodynamic roughness length of crevassed tidewater glaciers from UAV mapping**

*by Armin Dachauer, Richard Hann and Andrew J. Hodson*

**General comments**

This preprint addresses the important issue of the unknown aerodynamic roughness of inaccessible, heavily crevassed, tidewater glaciers. In this preprint, UAVs are used to map at high-resolution the elevation of the terminus of four tidewater glaciers in Svalbard. Five different methods, all based on the semi-empirical equation from Lettau (1969), are then used to map the aerodynamic roughness length for momentum $z_0$, thereby quantifying the large spatial variability of $z_0$ over these glaciers. Different sensitivity experiments are done, which confirm how much the modelled $z_0$ depends on the chosen elevation grid, but also on the wind direction.

This is well timed research, as atmospheric models have increasingly higher resolution and start resolving smaller parts of a glacier or ice sheet, including the very rough terminus of marine terminating glaciers. This research is also relevant, as the very rough nature of these surfaces is expected to increase turbulent heat fluxes and therefore runoff compared to smoother surfaces. The novelty in this preprint lies in the fact that multiple existing methods are compared over four new areas.

Overall, this preprint is well written and follows a clear and logical structure. Furthermore, the UAV digital elevation models (DEMs) are of high quality. The results section is interesting and the discussion addresses many uncertainties in this field. Nevertheless, the preprint can be improved further by clarifying several statements. Moreover, an important shortcoming of this work resides in the choice of the drag model to estimate $z_0$ from the measured DEMs. The associated (potentially large) errors should be addressed in more detail, as there is no in situ data to compare the model with. Finally, some parts of the discussion could be removed and/or shortened to make it more comprehensible. Therefore, I recommend publication after revision.

**Specific comments**

While the choice of using UAV DEMs with a grid size of 50m and an input resolution of 25cm/pixel is motivated, the choice of the semi-empirical equation

by Lettau (1969) to estimate $z_0$ is not clearly motivated. This equation relies on important assumptions, such as the absence of a displacement height and of a roughness sublayer, and the neglect of inter-obstacle sheltering. There is no reason to believe that these simplifications hold for such a complex, urban-like, surface. Besides, there is no mention of the typical turbulent flux fetch footprint, or about the value for the drag coefficient $c_d = 0.5$, which are both known to greatly influence the modelled $z_0$.

It may be argued that it is outside of the scope of this research to improve this model, yet it is important to know why the Lettau (1969) model was used over more recent models. One of these models is also mentioned in the preprint (Macdonald et al, 1998). This is even more relevant due to the fact that the performance cannot be assessed with in situ data, such as wind profile of turbulent flux measurements.

Other comments :

1. L16: replace "this heat exchange" by "the radiative heat fluxes", at least if this is what is meant here.

2. L21: the statement "it is a constant surface characteristic" is not clear, and seems to contradict the main conclusion of the preprint. Do the authors refer to the fact that the aerodynamic roughness length is often taken as a constant in atmospheric models ? Or that it does not depend on meteorological quantities ?

3. L23: Please rephrase "following the bulk approach". In its current form this statement may be confused with the bulk approach to estimate turbulent fluxes. I believe the authors refer to a different bulk approach.

4. L27 (& L3): Consider using "uncrewed" instead of "unmanned". Or remotely piloted aircraft system (RPAS).

5. L28: Please clarify how UAV overcomes the spatial coverage limitation of LiDAR. Aren't UAVs also limited in the area they cover ?

6. L35: Consider referring to the recent work by Van Tiggelen et al (Cryosph Discuss 2021, https://doi.org/10.5194/tc-2020-378),. They give estimated $z_0$ values for very rough ice & crevassed areas in west Greenland in their Figures 9 & 10.

7. L37: Consider rephrasing "makes it hard".

8. L87: the thesis of A. Dachauer could not be found online. Consider adding a public link with DOI to this reference.

9. L90: It is already assumed here that the mean wind direction coincides with the mean glacier slope, while this is only explicitly written at L95. Consider moving L95 before L90 for clarity.

10. L96: Please explain why assuming that the mean airflow is parallel to the slope means that the aerodynamic roughness is less influenced by the small-scale roughness features.

11. L99: Please clarify what is meant by "since small-scale roughness elements do not represent the real topographic expression".

12. L112: "[...] all four wind directions [...]".

13. L117: Is the value for $c_d = 0.5$ from Lettau (1969) realistic for crevasses ?

14. L119: Please rephrase "turns out". What do the authors exactly mean here ?

15. L127: Are the statistics immediately calculated on the transects taken from the detrended sub-grid ? Or are all the individual transects detrended once again ?

16. L134-136: I would argue that these statements are true only if the Lettau(1969) formula is used. More sophisticated models can be applied to a detrended profile that do take into account sheltering and obstacles of different spacing or height.

17. L144: Which parameters are calculated row-wise in the first two raster models, besides $h^*$ ? Possibly refer to Table 3 for clarity.

18. L157 (and L6) : It is not clear here whether $z_0$ varies by three or by four orders of magnitude. Please specify what is meant by "up to three (locally even four)".

19. L158: "The highest values": what are these values ?

20. L167: do the authors mean averaged over all four cardinal wind directions ? Or has the data been rotated over all 360 degrees ?

21. L171-172: refer to Table 4.

22. L178: At this point it might be useful (for future studies) to give a (short) interpretation on why the transect method yields a significantly larger $z_0$ than the raster method. Also see comment below about L322.

23. L186: This is hard to see on Fig5 with the given color scale. Consider changing the colormap or adding annotations in Fig5.

24. L190: Are the results shown in Fig5 different for wind blowing from the right than for wind blowing from the left ? If so, it would be useful to explain why. If not, consider removing panels c) and d).

25. Fig6: Please add to the caption what is denoted by the vertical extent of the boxplots (standard deviation, or quantiles ?). Consider using a logarithmic y axis, as all the means/medians seem clustered near y = 0m.

26. L201: Consider rephrasing "all the observed patterns recognized in the investigations of this study" by "all the patterns found in this study".

27. L207: Please clarify here that the resampling is only done in the following part. Otherwise the very different values for 2019 and 2020 in Table 4 do not make sense.

28. L213: The two sentences at L213-215 could be simplified in something like: "Although the deviations in $z_0$ are small, the lower values found in 2020 could be related to the fact ...".

29. L215: Would it be possible to check this in the true-color UAV images ? If so, this statement would be a very interesting example of how $z_0$ can rapidly change in time as well.

30. L245: A link could be made here with the assumptions of the Lettau (1969) model that does not account for the displacement height (or penetration depth). Underestimating crevasse depth using UAV could have a compensating effect on the modelled $z_0$.

31. L274: consider removing "the theory of".

32. L278: A grain roughness of 50 m is counter-intuitive. Does form drag not occur at scales smaller than 50 m ?

33. L280: What is a "considerable spatial resolution " ? Consider rephrasing in something like "50 m was chosen as it is the highest resolution that still includes the size of an average obstacle".

34. L282: consider renaming section "Model outputs of aerodynamic roughness length estimation" to "Estimated aerodynamic roughness length".

35. Figure 10: How do the authors know that parallel winds are more likely to occur or not than perpendicular winds ? Could it be that there is some confusion here in the interpretation of $\Omega$ ?

36. L332: Please explain (in the methods) how the raster methods take into account sheltering. Around L150 would be a good place.

37. L322-L332: these statements mostly repeat previous statements, so they could be removed. Subsection 4.3.1 could then be removed if lines L332-335 are added after L178.

38. L343: While it is true that Macdonald et al (1998) state that inter-obstacle sheltering becomes important at roughness densities above 20-30 %, they also show that the displacement height is non-negligible at roughness densities below 20 % (see their Fig.4). The latter is not taken into account in Lettau (1969).

39. L347: A roughness density of less than 0.10 - 0.15 is not the only criteria required for the equation by Lettau (1969) to be valid. Consider also replacing "this study shows" by "this study assumes".

40. L350 Perhaps section 4.3.3 can be made more compact. In its current form it mostly repeats previous research with generic statements.

41. L364-371: I propose to remove this subsection. Without a direct comparison with wind profiles or turbulent fluxes, the statement that one model performs better over another is difficult, if not impossible, to make. Instead a discussion, or possibly a sensitivity analysis to quantify model uncertainty (relating to the equation from Lettau) would be beneficial.

42. L366: I would argue that the raster methods are also based on the same assumptions, as they are all based on Lettau (1969).

43. L389: The statement "leading to a better representation of turbulent heat fluxes" was not proven in this preprint. I suggest rephrasing this in something like "potentially leading to a better representation of turbulent heat fluxes". This prevents any confusion when only reading the conclusion.

**Technical corrections**

1. L173: "for heavily crevassed areas."

2. L213: "This small deviation in z0 values makes ..."

3. L217 (and Fig9): "independently"

4. L284: "obtain" to "contain"

5. L292: "methods"

6. L294: "... still mostly positive ratio values, it is the glacier ..."

---

## Referee Comment (RC2)

Review of 'Aerodynamic roughness length…' submitted by Dachauer et al to *The Cryosphere*

The study by Dachauer et al utilizes UAV-derived digital elevation models to estimate the surface roughness (specifically, the aerodynamic roughness length) for the heavily-crevassed terminus area of four glaciers in Svalbard. This is an interesting and practical application of UAV data to a domain where field measurements are dangerous, if not impossible, so few estimates and no direct measurements of roughness are yet available for heavily crevassed ice. The authors utilize five contending approaches to estimating $z_0$ from DEMS, and show that all approaches represent similar spatial variability across the study domains, and that some approaches show little scale dependence when an appropriate grid scale is used. For all methods, $z_0$ values in crevassed zones are considerably higher than values typically used for glacier ice, as expected. Overall this is a nice study, demonstrating the approximate range of roughness values to consider for these hard-to-reach parts of glaciers, but its impact may suffer from the difficulty in constraining or validating the models. Still, with some additional analyses to understand the importance of uncertainty in modelled $z_0$, the manuscript will be a nice contribution to the literature. I have a number of relatively minor suggestions for the authors to consider in preparing a revision.

Main comments.

1. The fact that no ground control was used in the study should be clearly mentioned before the discussion. Certainly it would be difficult to constrain the glacierized portion of the DEMs, but some of the exposed bedrock around the glaciers' lateral margins could have reduced the positional errors. In addition, many drones now have RTK-GPS modules that are perfectly suited to the no-GCP application, and can achieve centimetre-accuracy (see the Chudley et al, paper; others are also available). I also think that the GPS accuracy values from the manufacturer are probably not reliable, at least in the Z direction (however the UAV also uses pressure sensors that may improve the relative altitude precision). At first glance this seemed to be a major limitation for the study, as poor georeferencing control can lead to DEM warping, as well as shifts. However, I actually think this should be a minor problem for your application, which focuses on the relative differences in surface topography, in part due to the detrending you have applied. In short, in the methods I would suggest that you acknowledge the challenges of establishing precise GCP controls for this type of situation, as well as that newer platforms mitigate these issues, but assert that this will not be a problem for your specific application

2. The authors have reoriented their DEMs to the dominant direction of glacier flow (at the terminus) to analyse roughness in the down-glacier and cross-glacier directions. I would encourage the authors to consider multi-directional wind patterns more carefully in the discussion. You might reframe some of the anisotropy discussion to consider that the down- and cross- glacier roughness estimates provide end-members for the temporally variable roughness experienced at a site on these glaciers.

3. A key limitation for this study (and the possible advantage to using UAVs) is the inability to validate the estimates, which are well correlated to one another but differ in magnitude. This has been a problem for other similar efforts, even with local measurements (e.g. Miles et al, 2017), although there is promise to reduce the scale dependence (e.g. Chambers et al, 2021). For your application, the question is – how believable are the heavily-crevassed-area roughness values? This is very difficult to pin down, but I think the five methods tested provide constraint to within an order of magnitude. Maybe you can evaluate your estimates for the smoother area of some glaciers to refine this range further, but I think an important question is also – how precise do models need $z_0$ to be prescribed? I think this is an

important discussion topic for the manuscript, since you cannot constrain the values precisely; how different might turbulent fluxes be in crevassed areas considering the range of values? Is an unconstrained estimate already 'good enough' and not likely to change the results, or do we need to determine the accuracy precisely?

4. Related to the above, a very nice possible outcome would be to consider a reduced-complexity parameterization of roughness for crevassed areas. This could be related to crevasse spacing (or possibly depth) or a damage factor. I suggest this because 1) crevasses can be readily mapped using high-resolution satellite imagery (Pleiades or even PlanetScope), and 2) models of glacier dynamics are inceasingly able to resolve complex stress and strain patterns near glacier termini, and this could enable an improved link to surface energy balance. It is also worth noting that unlike most glacier surfaces, the roughness of a heavily crevassed glacier terminus is not likely to see much seasonal change, as the dominant roughness elements are unaffected by snow/ice melt processes.

Minor comments.

L7. 'best accounts' – as formulated, this sentence is a bit misleading, as you are not demonstrating that the moving-window approach is worse than the sub-grid approach (the strict interpretation of the sentence as written). Rather, your results indicate that 50m is a suitable distance for detrending, since the scale dependence for most approaches breaks down above that distance.

L16. Suggest 'balance' to 'lead to'

L21. I'd recommend reformulating this sentence. $z_0$ is certainly *not* a constant surface characteristic. At the very least it changes considerably in time (e.g. Brock et al, 2006; Smeets studies), but of course turbulence is not only affected by the surface itself, but the wind speed and direction.

L37. I don't recall Quincey et al (2017) looking at crevasses?

L88. Please add some details of the final configuration here.

L89. The lack of ground control points should be addressed directly here.

Figure 3. Could you add depiction of the moving-window formulation (e.g. Fitzpatrick et al, 2019)? The detrending approach is indeed quite crucial for obstacle definition.

L113. 'all wind directions' – by this you mean the four directions of the coordinate system defined in Fig 2b, and not every 15degree increment, for example

L120. 'a lot' appears in the text 'a lot'. Please consider a less colloquial formulation

L127. For consistency with prior descriptions, please indicate 'up-crossing' somehow here.

L132. This adaptation need to be established a bit more carefully. The use of cross-profile (instead of along-profile) obstacles dates back to Lettau (1969) if not before. The rationale is that if the bumps resolved in this matter equate to the silhouette area facing the wind. This is true when surfaces do not have a clear grain, in which case you get channelized flow rather than turbulence. Note that Munro also found a 4x difference based on transect direction for ablating ice that showed a clear grain (similar to your own magnitude of differences). I actually agree with this profile rotation

considering the strong grain of the surface (since you have also considering skimming flow!), but some additional justification is needed in the text.

L137. The precise implementation of the 'transect' approaches is not clear. Do you determine a $z_0$ value for each transect, then combine them (and how)? Or do you accumulate obstacles from all transects (as in Miles et al (2017))?

L180. Technically, these are not 'cardinal' wind directions.

L194-5. Which are the 'both parameters'? Not clear.

Figure 5. Please increase the font size for all axes and the colorbar.

Figure 6. Please use a log scale for the y-axis.

Figure 7. Please increase the font size for all axes and the colorbar.

L202. The similarity of values between glaciers (for the crevassed areas) raises an interesting question – can you parameterize $z_0$ based on crevasse density directly? If so, this would be a promising avenue to estimate $z_0$ (at least for the crevassed areas) without needing high-resolution DEMs.

L212. It's nice to be able to compare two different years. This similarity is also not so surprising since the roughness elements (crevasses) are probably not as transient as for other glacier surfaces.

L214. This sentence about the decrease does not make sense – do you think the 10% reduction is meaningful? I would be very sceptical.

Figure 8. Please eliminate the duplicate colorbar

Figure 9. I would again recommend plotting this with a logarithmic y-axis. Please also annotate the median obstacle sizes determined for each site (then the 50m grid size used).

L241. The nadir views at interval probably do not not resolve topography very far into crevasses.

L252. I believe you are arguing that you need a DEM with high precision (rather than high accuracy). I would agree (for a microtopographic $z_0$ calculation) as measuring the local features is more crucial than the elevation of those features. The manufacturer hover accuracy estimates are not terribly relevant for your purpose, though – I would suggest that this is the best-case accuracy. Without GCPs, DEM warping can be particularly problematic.

L256 – L262. It is not clear how you measure % distortion in this context. % of what?

L279-L280. This is a nice theoretical justification of the choice of grid size, but 50m is still quite arbitrary, looking at Figure 9, which does not show a kink but a smooth progression that might just be an artefact of the logarithmic scale of $z_0$ variability. I think you could provide better evaluation of this choice of grid size – for example, what range of $z_0$ values does this give for the smoother domains, and how does that correspond to expected values? If in fact the transition from grain to form roughness occurs at 30m (or 70m) how much are the distributed roughness estimates changed?

L298-299. Without a doubt, katabatic winds are a predominant wind direction for mountain glaciers, and are also important for tidewater glaciers, but external forcing also plays a role, especially for the latter group. The key drivers leading to their formation (altitudinal/temperature gradients, combined with topographic chanelling) are weaker at the polar tidewater sites studied here. There are some interesting results related to this from e.g. Esau and Repina (2012). Multidirectional wind speed can play an important role in temporal variations in $z_0$, and although you don't need to go so far as to consider turbulence footprints here (Steiner et al, 2018; Nicholson and Stiperski, 2020), I think it is useful to note that the cross- and down- glacier roughness elements both influence the effective $z_0$ at a site.

L305. The validation is a challenge for this study. I wonder if you can consider real-world analogues to crevassed areas (outside of glaciology) that might have been investigated previously. Lettau (1969) included a variety of other surfaces (even including urban areas) that could be relevant reference points to check the order of magnitude.

L397. I'd recommend making your DEMs and code publicly archived.

Brock, B. W., Willis, I. C., & Sharp, M. J. (2006). Measurement and parameterization of aerodynamic roughness length variations at Haut Glacier d'Arolla, Switzerland. *Journal of Glaciology*, 52(177), 281–297. https://doi.org/10.3189/172756506781828746

Chambers, J. R., Smith, M. W., Smith, T., Sailer, R., Quincey, D. J., Carrivick, J. L., … James, M. R. (2021). Correcting for Systematic Underestimation of Topographic Glacier Aerodynamic Roughness Values From Hintereisferner, Austria. *Frontiers in Earth Science*, 9(May), 1–16. https://doi.org/10.3389/feart.2021.691195

Fitzpatrick, N., Radic, V., & Menounos, B. (2019). A multi-season investigation of glacier surface roughness lengths through in situ and remote observation. *The Cryosphere*, 13, 1051–1071. https://doi.org/10.5194/tc-13-1051-2019

Igor Esau, Irina Repina, "Wind Climate in Kongsfjorden, Svalbard, and Attribution of Leading Wind Driving Mechanisms through Turbulence-Resolving Simulations", *Advances in Meteorology*, vol. 2012, Article ID 568454, 16 pages, 2012. https://doi.org/10.1155/2012/568454

Lettau, H. H. (1969). Note on Aerodynamic Roughness-Parameter Estimation on the Basis of Roughness-Element Description.pdf. *Journal of Applied Meteorology*, 8, 828–832. https://doi.org/10.1175/1520-0450(1969)008%3C0828:NOARPE%3E2.0.CO;2

Miles, E. S., Steiner, J. F., & Brun, F. (2017). Highly variable aerodynamic roughness length ( $z_0$ ) for a hummocky debris-covered glacier. *Journal of Geophysical Research: Atmospheres*. https://doi.org/10.1002/2017JD026510

Nicholson, L., & Stiperski, I. (2020). Comparison of turbulent structures and energy fluxes over exposed and debris-covered glacier ice. *Journal of Glaciology*, 66(258), 543–555. https://doi.org/https://doi.org/10.1017/jog.2020.23

Quincey, D. J., Smith, M. W., Rounce, D. R., Ross, A., King, O., & Watson, C. S. (2017). Evaluating morphological estimates of the aerodynamic roughness of debris covered glacier ice. *Earth Surface Processes and Landforms*, 42, 2541–2553. https://doi.org/10.1002/esp.4198

Smeets, C. J. P. P., Duynkerke, P. G., & Vugts, H. F. (1999). Observed wind profiles and turbulent fluxes over an ice surface with changing surface roughness. *Boundary-Layer Meteorology*, 92(1), 99–121. https://doi.org/10.1023/A:1001899015849

Smeets, C. J. P. P., & Broeke, M. R. (2008). Temporal and spatial variations of the aerodynamic roughness length in the ablation zone of the greenland ice sheet. *Boundary-Layer Meteorology*, 128(3), 315–338. https://doi.org/10.1007/s10546-008-9291-0

Steiner, J. F., Litt, M., Stigter, E. E., Shea, J., Bierkens, M. F. P., & Immerzeel, W. W. (2018). The Importance of Turbulent Fluxes in the Surface Energy Balance of a Debris-Covered Glacier in the Himalayas. *Frontiers in Earth Science*, 6(October), 1–18. https://doi.org/10.3389/feart.2018.00144

---

## Author Comment (AC1)

**Reply to review comments by Maurice Van Tiggelen**

September 20, 2021

**General comments**

*This preprint addresses the important issue of the unknown aerodynamic roughness of inaccessible, heavily crevassed, tidewater glaciers. In this preprint, UAVs are used to map at high-resolution the elevation of the terminus of four tidewater glaciers in Svalbard. Five different methods, all based on the semi-empirical equation from Lettau (1969), are then used to map the aerodynamic roughness length for momentum $z_0$, thereby quantifying the large spatial variability of $z_0$ over these glaciers. Different sensitivity experiments are done, which confirm how much the modelled $z_0$ depends on the chosen elevation grid, but also on the wind direction. This is well timed research, as atmospheric models have increasingly higher resolution and start resolving smaller parts of a glacier or ice sheet, including the very rough terminus of marine terminating glaciers. This research is also relevant, as the very rough nature of these surfaces is expected to increase turbulent heat fluxes and therefore runoff compared to smoother surfaces. The novelty in this preprint lies in the fact that multiple existing methods are compared over four new areas. Overall, this preprint is well written and follows a clear and logical structure. Furthermore, the UAV digital elevation models (DEMs) are of high quality. The results section is interesting and the discussion addresses many uncertainties in this field. Nevertheless, the preprint can be improved further by clarifying several statements. Moreover, an important shortcoming of this work resides in the choice of the drag model to estimate $z_0$ from the measured DEMs. The associated (potentially large) errors should be addressed in more detail, as there is no in-situ data to compare the model with. Finally, some parts of the discussion could be removed and/or shortened to make it more comprehensible. Therefore, I recommend publication after revision.*

Reply: The authors highly appreciate the constructive and thoughtful comments. In the text following below the referee's comments are written in *italic* and the line numbers refer to the original review version of the manuscript, if not specifically mentioned otherwise.

**Specific comments**

*While the choice of using UAV DEMs with a grid size of 50m and an input resolution of 25cm/pixel is motivated, the choice of the semi-empirical equation by Lettau (1969) to estimate $z_0$ is not clearly motivated. This equation relies on important assumptions, such as the absence of a displacement height and of a roughness sublayer, and the neglect of inter-obstacle sheltering. There is no reason to believe that these simplifications hold for such a complex, urban-like, surface. Besides, there is no mention of the typical turbulent flux fetch footprint, or about the value for the drag coefficient $c_d = 0.5$, which are both known to greatly influence the modelled $z_0$.*
*It may be argued that it is outside of the scope of this research to improve this model, yet it is important to know why the Lettau (1969) model was used over more recent models. One of these models is also mentioned in the preprint (Macdonald et al, 1998). This is even more relevant due to the fact that the performance cannot be assessed with in situ data, such as wind profile of turbulent flux measurements.*

Reply: The authors agree with the referee that the Lettau (1969) model is based on many assumptions which not necessarily hold for complex surfaces, such as highly crevassed tidewater glaciers. However, we would argue that the choice of the five models is exactly motivated to face this shortcoming. While

the two transect models are based on many assumptions (namely roughness elements are of equal height, uniformly distributed, isotropic and not affected by any sheltering), the three raster method models do not rely on these assumptions since the frontal area $s$ can be calculated directly (Smith et al., 2016). Therefore, the choice of the five models can be justified in a way that they highlight the influence of these assumptions and further show the impact of other parameters such as $h^*$ on the $z_0$ estimates. For clarification, this justification was added to the methods part of the manuscript. Additionally, the errors/shortcomings associated to each model and variable definition was further elaborated in the discussion of the manuscript. The ground area $S$ which corresponds to the chosen grid size and the according profile length is a simple approximation to the real fetch footprint. The justification of the grid size was discussed extensively from L97 onwards of the updated manuscript version. However, the simplification allows us to estimate $z_0$ values for the four wind directions and additionally provides uniform parameterization throughout all glaciers and models (this justification was added to the paper discussion). We find the proposed concerns coming along with the choice of the drag coefficient to be reasonable. However, for this study we decided to stick to the in glaciology commonly employed drag coefficient definition of $c_d$=0.5 and to discuss the shortcoming of this assumption in the discussion part of the study.

**Other comments:**

1. L16: *replace "this heat exchange" by "the radiative heat fluxes", at least if this is what is meant here.*
   Reply: The sentence was rephrased as follows: "Both sensible and latent heat fluxes balance lead to this heat exchange on the surface and therefore have a large impact on the meltwater production and the surface energy balance of glaciers."

2. L21: *the statement "it is a constant surface characteristic" is not clear, and seems to contradict the main conclusion of the preprint. Do the authors refer to the fact that the aerodynamic roughness length is often taken as a constant in atmospheric models? Or that it does not depend on meteorological quantities?*
   Reply: We refer to the fact that $z_0$ is independent of any meteorological quantities. An according sentence was added: "It is a surface characteristics and therefore independent of meteorological quantities".

3. L23: *Please rephrase "following the bulk approach". In its current form this statement may be confused with the bulk approach to estimate turbulent fluxes. I believe the authors refer to a different bulk approach.*
   Reply: The authors refer to the bulk approach to estimate turbulent fluxes. For clarification, L24 was adjusted to: "The bulk approach for the calculation of those turbulent fluxes is very popular...". Additionally, "following the bulk approach" was changed into "have determined $z_0$ values of glacier surfaces based on the bulk approach".

4. L27 (& L3): *Consider using "uncrewed" instead of "unmanned". Or remotely piloted aircraft system (RPAS).*
   Reply: The authors very much acknowledge the point of that comment and the importance behind it. However, since the term 'unmanned' is the official technical term used for such vehicles (e.g. in regulations), we decided to use it. Nevertheless, we added a sentence defining other used terms: "often also called uncrewed vehicle systems or drones" to acknowledge the issue.

5. L28: *Please clarify how UAV overcomes the spatial coverage limitation of LiDAR. Aren't UAVs also limited in the area they cover?*
   Reply: For clarification, the following sentence was added: "... since they are more flexible in their use and less limited by local topology as they provide a bird's-eye perspective."

6. L35: *Consider referring to the recent work by Van Tiggelen et al (CryosphDiscuss 2021, https://doi.org/10.5194/tc-2020-378). They give estimated z0 values for very rough ice & crevassed areas in west Greenland in their Figures 9 & 10.*

Reply: The 'typical values range for rough glacier ice' was adjusted and the suggested reference added. Furthermore, the study was mentioned in the opening sentence of that paragraph describing existing research.

7. L37: *Consider rephrasing "makes it hard".*
   Reply: "makes it hard to define" changed to "makes the definition of z0 values challenging"

8. L87: *The thesis of A. Dachauer could not be found online. Consider adding a public link with DOI to this reference.*
   Reply: A DOI of the thesis does not exist, but a link to the pdf file was added to the reference.

9. L90: *It is already assumed here that the mean wind direction coincides with the mean glacier slope, while this is only explicitly written at L95. Consider moving L95 before L90 for clarity.*
   Reply: The sentence was rephrased as follows to acknowledge the mentioned issue: "This allowed the estimation of $z_0$ values for the following four wind directions: down-glacier, up-glacier and cross-glacier from both sides"

10. L96: *Please explain why assuming that the mean airflow is parallel to the slope means that the aerodynamic roughness is less influenced by the small-scale roughness features.*
    Reply: The sentence was rephrased for clarification (see comment 11./L99). Additionally "means" was replaced by "justifies the assumption"

11. L99: *Please clarify what is meant by "since small-scale roughness elements do not represent the real topographic expression".*
    Reply: The following two sentences were added/rephrased to clarify the topic: "In other words, only looking at the small-scale surface roughness elements would lead to the wrong roughness parameterization, since they might be located on the inner side of a large-scale roughness obstacle not exposed to the whole mean airflow. Accordingly, the chosen grid size must be large enough to include the macro-structure of the surface, since small-scale roughness elements do not represent the real topographic expression."

12. L112: *"[...] all four wind directions [...]".*
    Reply: suggestion implemented.

13. L117: *Is the value for $c_d$ = 0.5 from Lettau (1969) realistic for crevasses?*
    Reply: This is indeed a broadly discussed issue and potentially leading to an impact on the $z_0$ results. However, we decided to stick to that widely adopted definition and added the following sentence on L356 of the updated manuscript version in the discussion to draw attention to this assumption: "Furthermore, the widely adopted drag coefficient of Lettau (1969) $c_d = 0.5$ corresponds to an average form drag effect on roughness elements. Its rationale is widely discussed since it does not necessarily hold for heterogeneous locations (Quincey et al., 2017)."

14. L119: *Please rephrase "turns out". What do the authors exactly mean here?*
    Reply: "turns out" changed to "is". Additionally, the sentence was rephrased to make it clearer: "roughness elements are non-uniform" was added.

15. L127: *Are the statistics immediately calculated on the transects taken from the detrended sub-grid? Or are all the individual transects detrended once again?*
    Reply: The detrending happened row-wise (e.g. for each individual transect) at the first place. The sentence was rephrased for clarification: from "Each row of the detrended sub-grid was treated..." to "Each row of the sub-grid was detrended and treated... "

16. L134-136: *I would argue that these statements are true only if the Lettau(1969) formula is used. More sophisticated models can be applied to a detrended profile that do take into account sheltering and obstacles of different spacing or height.*
    Reply: All models are based on the Lettau (1969) formula, but the raster methods were modified in a way that they take into account sheltering and different obstacle sizes (see reply to comment 42.)

17. L144: *Which parameters are calculated row-wise in the first two raster models, besides? Possibly refer to Table 3 for clarity.*
Reply: As already stated on L148, s and $h^*$ are calculated row-wise. A reference to Table 3 was added.

18. L157 (and L6) : *It is not clear here whether z0 varies by three or by four orders of magnitude. Please specify what is meant by "up to three (locally even four)".*
Reply: For one model the range of $z_0$ values is up to three orders of magnitude. Between different models this can of course add another order of magnitude. The authors agree that this creates confusion and therefore decided to remove the phrase "(locally even four)".

19. L158: *"The highest values": what are these values?*
Reply: "The highest values" corresponds to the larger values (dm-m scale) in the mentioned range of $z_0$ values. The following sentences were rephrased for clarification: "The largest $z_0$ values (decimeter to meter scale) were found close to the glacier front where crevasses are big and steep using the transect methods and for winds blowing parallel to the flow direction of the glacier. The lowest values (milimeter to centimeter scale) were estimated with the raster methods for smooth, crevasse-free ice and for cross-glacier wind directions."

20. L167: *do the authors mean averaged over all four cardinal wind directions? Or has the data been rotated over all 360 degrees?*
Reply: The authors mean averaged over all four cardinal wind directions. For clarification the sentence was rephrased to: "over all four wind directions".

21. L171-172: *refer to Table 4.*
Reply: Suggestion implemented.

22. L178: *At this point it might be useful (for future studies) to give a (short) interpretation on why the transect method yields a significantly larger z0 than the raster method. Also see comment below about L322.*
Reply: suggestion implemented.

23. L186: *This is hard to see on Fig5 with the given color scale. Consider changing the colormap or adding annotations in Fig5.*
Reply: For clarification, the text was rephrased as follows: "However, Fig. 5 illustrates that larger roughness elements, which can be found close to the glacier front, for instance, present a stronger wind dependency because they vary more strongly with changing wind directions (from dm to m scale) compared to areas that are less crevassed like on the upper part of Fridtjovbreen (similar $z_0$ values in mm scale for all wind directions)."

24. L190: *Are the results shown in Fig5 different for wind blowing from the right than for wind blowing from the left? If so, it would be useful to explain why. If not, consider removing panels c) and d).*
Reply: They are different but very similar. To clarify, the following sentence was added: "The two cross-glacier (up- and down-glacier, respectively) wind directions lead to very similar $z_0$ values since they are both calculated on the same transect but from opposing wind directions."

25. Fig6: *Please add to the caption what is denoted by the vertical extent of the boxplots (standard deviation, or quantiles?). Consider using algorithmic y axis, as all the means/medians seem clustered near y = 0m.*
Reply: The following sentence was added to the caption: "Whiskers are visualizing the variability outside the upper and lower quartiles up to 1.5 times the interquartile range.". Furthermore, the y-axis was changed to logarithmic.

26. L201: *Consider rephrasing "all the observed patterns recognized in the investigations of this study" by "all the patterns found in this study".*
Reply: Suggestion implemented.

27. L207: *Please clarify here that the resampling is only done in the following part. Otherwise the very different values for 2019 and 2020 in Table 4 do not make sense.*
Reply: To clarify this issue, the following sentence was added: "Furthermore, for this particular comparison [...]".

28. L213: *The two sentences at L213-215 could be simplified in something like: "Although the deviations in z0 are small, the lower values found in 2020 could be related to the fact ...".*
Reply: In consultation of the second referee's comment the authors decided to remove the whole sentence since the deviations were too small to give reason for a meaningful explanation.

29. L215: *Would it be possible to check this in the true-color UAV images? If so, this statement would be a very interesting example of how z0 can rapidly change in time as well.*
Reply: Potential snow-bridges could not be determined in the true-color UAV images of both years since it's hard to distinguish ice and snow especially in the deep crevasses. Nevertheless, there might still be some remaining snow-bridges deep in the crevasses. However, since the deviations are so small and the impact of snow-bridges deep inside the crevasses might be minor we decided to not further question the rationale of a 10 % deviation.

30. L245: *A link could be made here with the assumptions of the Lettau (1969) model that does not account for the displacement height (or penetration depth). Underestimating crevasse depth using UAV could have a compensating effect on the modelled z0.*
Reply: The following text was added to address this issue: "Equation 1 of Lettau (1969) and the transect methods are not defining any penetration depth limit. The raster method however, assumes that effective roughness only depends on the roughness elements above the detrended plane level which indicates how far the effective turbulent mixing advances into the crevasses."

31. L274: *consider removing "the theory of".*
Reply: Suggestion implemented.

32. L278: *A grain roughness of 50 m is counter-intuitive. Does form drag not occur at scales smaller than 50 m?*
Reply: For clarification, the following sentence was added/rephrased: "Accordingly, given the large roughness elements investigated in our study, the 'grain' roughness is assumed to belong to the texture on the crevasses. "

33. L280: *What is a "considerable spatial resolution "? Consider rephrasing in something like "50 m was chosen as it is the highest resolution that still includes the size of an average obstacle".*
Reply: The sentence was rephrased to: "50 m was chosen, since it is the smallest resolution possible to still include the size of an average obstacle."

34. L282: *consider renaming section "Model outputs of aerodynamic roughness length estimation" to "Estimated aerodynamic roughness length".*
Reply: Suggestion implemented.

35. Figure 10: *How do the authors know that parallel winds are more likely to occur or not than perpendicular winds? Could it be that there is some confusion here in the interpretation of $\Omega$?*
Reply: Several studies (e.g. Fitzpatrick et al., 2019; Karner et al., 2013) found the winds to be more likely to blow parallel, due to katabatically forced down-slope winds. However, this is not related directly to the results of Figure 10. Figure 10 only compares the $z_0$ values of parallel vs. perpendicular flowing winds showing that $z_0$ values are strongly anisotropic and parallel winds are dominant.

36. L332: *Please explain (in the methods) how the raster methods take into account sheltering. Around L150 would be a good place.*
Reply: In the methods part (line 141) the following sentence was added: "In the raster method all areas below the detrended plane were neglected, assuming that they would be effectively sheltered."

37. L322-L332: *these statements mostly repeat previous statements, so they could be removed. Subsection 4.3.1 could then be removed if lines L332-335 are added after L178.*
Reply: suggestion implemented: L332-335 were added after L178 and L322-L332 were deleted. Additionally, the following sentence: "This indicates that the $z_0$ values are correlated, but does not provide any conclusion about the quality of the individual models." was added after L169.

38. L343: *While it is true that Macdonald et al (1998) state that inter-obstacle sheltering becomes important at roughness densities above 20-30 %, they also show that the displacement height is non-negligible at roughness densities below 20 % (see their Fig.4). The latter is not taken into account in Lettau (1969).*
Reply: The authors agree that Lettau (1969) and accordingly the two transect method models, are not taking into account the displacement height. However, the three raster method models are assuming the displacement height to correspond with the detrended plane. Therefore, a comparison between the different models gives insights in the impact of a considered displacement height.

39. L347: *A roughness density of less than 0.10 - 0.15 is not the only criteria required for the equation by Lettau (1969) to be valid. Consider also replacing "this study shows" by "this study assumes".*
Reply: For clarification, the words "with respect to the sheltering effect" were added. Additionally, "this study assumes" was implemented.

40. L350: *Perhaps section 4.3.3 can be made more compact. In its current form it mostly repeats previous research with generic statements.*
Reply: Section 4.3.3 was rephrased into a more compact version. Additionally, the subdivision into subsections was removed.

41. L364-371: *I propose to remove this subsection. Without a direct comparison with wind profiles or turbulent fluxes, the statement that one model performs better over another is difficult, if not impossible, to make. Instead a discussion, or possibly a sensitivity analysis to quantify model uncertainty (relating to the equation from Lettau) would be beneficial.*
Reply: the mentioned section was removed. Additionally, to further evaluate the model uncertainty the impact of $h^*$, $s$ and $c_d$ was discussed in the prior subsection.

42. L366: *I would argue that the raster methods are also based on the same assumptions, as they are all based on Lettau (1969).*
Reply: The raster method is not based on these assumptions since the raster methods which are based on Lettau (1969) were further modified to exactly prevent these shortcomings (see e.g. Smith et al., 2016).

43. L389: *The statement "leading to a better representation of turbulent heat fluxes" was not proven in this preprint. I suggest rephrasing this in something like "potentially leading to a better representation of turbulent heat fluxes". This prevents any confusion when only reading the conclusion.*
Reply: Suggestion implemented.

**Technical corrections**

1. L173: *"for heavily crevassed areas."*
Reply: Suggestion implemented and "up to" added.

2. L213: *"This small deviation in z0 values makes ..."*
Reply: Sentence was rephrased according to Editor's comment 28 in section "Specific comments".

3. L217 (and Fig9): *"independently"*
Reply: Suggestion implemented.

4. L284: *"obtain" to "contain"*
Reply: Suggestion implemented.

5. L292: *"methods"*
   Reply: Suggestion implemented.

6. L294: *"... still mostly positive ratio values, it is the glacier ..."*
   Reply: Suggestion implemented.

**References:**
Fitzpatrick, N., Radi´c, V., and Menounos, B.: A multi-season investigation of glacier surface roughness lengths through in situ and remote observation, The Cryosphere, 13, 1051–1071, https://doi.org/10.5194/tc-13-1051-2019, 2019.

Karner, F., Obleitner, F., Krismer, T., Kohler, J., and Greuell,W.: A decade of energy and mass balance investigations on the glacier Kongsvegen, Svalbard, Journal of Geophysical Research Atmospheres, 118, 3986–4000, https://doi.org/10.1029/2012JD018342, 2013.
Lettau, H.: Note on aerodynamic roughness-parameter estimation on the basis of roughness-element description, Journal of applied meteorology, 8, 828–832, 1969.

Quincey, D., Smith, M., Rounce, D., Ross, A., King, O., and Watson, C.: Evaluating morphological estimates of the aerodynamic roughness of debris covered glacier ice, Earth Surface Processes and Landforms, 42, 2541–2553, https://doi.org/10.1002/esp.4198, 2017.

Smith, M. W., Quincey, D. J., Dixon, T., Bingham, R. G., Carrivick, J. L., Irvine-Fynn, T. D. L., and Rippin, D. M.: Aerodynamic roughness of glacial ice surfaces derived from high-resolution topographic data, Journal of Geophysical Research: Earth Surface, 121, 748–766, https://doi.org/10.1002/2015JF003759, 2016.

---

## Author Comment (AC2)

**Reply to review comments by Evan Miles**

September 20, 2021

*The study by Dachauer et al utilizes UAV-derived digital elevation models to estimate the surface roughness (specifically, the aerodynamic roughness length) for the heavily-crevassed terminus area of four glaciers in Svalbard. This is an interesting and practical application of UAV data to a domain where field measurements are dangerous, if not impossible, so few estimates and no direct measurements of roughness are yet available for heavily crevassed ice. The authors utilize five contending approaches to estimating $z_0$ from DEMS, and show that all approaches represent similar spatial variability across the study domains, and that some approaches show little scale dependence when an appropriate grid scale is used. For all methods, $z_0$ values in crevassed zones are considerably higher than values typically used for glacier ice, as expected. Overall this is a nice study, demonstrating the approximate range of roughness values to consider for these hard-to-reach parts of glaciers, but its impact may suffer from the difficulty in constraining or validating the models. Still, with some additional analyses to understand the importance of uncertainty in modelled $z_0$, the manuscript will be a nice contribution to the literature. I have a number of relatively minor suggestions for the authors to consider in preparing a revision.*

Reply: The authors are thankful to the referee for the precise and thoughtful comments. In the following text the comments of the referee, are written in *italic* and the line numbers refer to the original review version of the manuscript, if not specifically mentioned otherwise.

**Main comments**

1. *The fact that no ground control was used in the study should be clearly mentioned before the discussion. Certainly it would be difficult to constrain the glacierized portion of the DEMs, but some of the exposed bedrock around the glaciers' lateral margins could have reduced the positional errors. In addition, many drones now have RTK-GPS modules that are perfectly suited to the no-GCP application, and can achieve centimetre-accuracy (see the Chudley et al, paper; others are also available). I also think that the GPS accuracy values from the manufacturer are probably not reliable, at least in the Z direction (however the UAV also uses pressure sensors that may improve the relative altitude precision). At first glance this seemed to be a major limitation for the study, as poor georeferencing control can lead to DEM warping, as well as shifts. However, I actually think this should be a minor problem for your application, which focuses on the relative differences in surface topography, in part due to the detrending you have applied. In short, in the methods I would suggest that you acknowledge the challenges of establishing precise GCP controls for this type of situation, as well as that newer platforms mitigate these issues, but assert that this will not be a problem for your specific application*

   Reply: The authors agree with the comment of the referee. The lack of GCP control points as well as its minor impact on the results is now mentioned in the methods part (see L91 of updated manuscript). Additionally, alternative platforms such as RTK-corrections (Chudley et al., 2019) are also mentioned: "Due to an inaccessible glacier surface, no ground control points (GCPs) could be placed on the mapped area for georeferencing. No alternative georeferencing platforms such as real-time-kinematic correction (Chudley et al., 2019) were available. While this is a recommended procedure for future application of this technique, we point out that computation of $z_0$ requires quantification of relative topographic differences and so the impact of this shortcoming is minor".

2. *The authors have reoriented their DEMs to the dominant direction of glacier flow (at the terminus) to analyse roughness in the down-glacier and cross-glacier directions. I would encourage the authors to consider multi-directional wind patterns more carefully in the discussion. You might reframe some of the anisotropy discussion to consider that the down-and cross-glacier roughness estimates provide end-members for the temporally variable roughness experienced at a site on these glaciers.*

    Reply: To address the mentioned concern, some parts of the anisotropy discussion were rephrased. In more detail, the strong focus on down-glacier wind direction was lifted by considering the study of Esau and Repina (2012). Additionally, the temporal variability of $z_0$ due to changing wind directions was addressed in the discussion: "However, Esau and Repina (2012) found the katabatically forced wind systems to be of less significance for tidewater glaciers, highlighting the influence of wind direction on effective $z_0$ values. Furthermore, the wind direction dependency indicates the temporal variability of the aerodynamic roughness length due to changing wind directions from daily up to seasonal time scales.".

3. *A key limitation for this study (and the possible advantage to using UAVs) is the inability to validate the estimates, which are well correlated to one another but differ in magnitude. This has been a problem for other similar efforts, even with local measurements (e.g. Miles et al, 2017), although there is promise to reduce the scale dependence (e.g. Chambers et al, 2021). For your application, the question is –how believable are the heavily-crevassed-area roughness values? This is very difficult to pin down, but I think the five methods tested provide constraint to within an order of magnitude. Maybe you can evaluate your estimates for the smoother area of some glaciers to refine this range further, but I think an important question is also –how precise do models need z0 to be prescribed? I think this is an important discussion topic for the manuscript, since you cannot constrain the values precisely; how different might turbulent fluxes be in crevassed areas considering the range of values? Is an unconstrained estimate already 'good enough' and not likely to change the results, or do we need to determine the accuracy precisely?*

    Reply: The authors agree with the referee and decided to include the discussion of the required $z_0$ estimation accuracy for models into the manuscript. Therefore, the following text was added in the discussion: "The validation of the model estimates remains a big challenge due to the lack of reference values. Thus, it should be questioned whether the modelled $z_0$ range of about an order of magnitude is accurate enough for energy balance models. In general, an increase of $z_0$ by one order of magnitude will more than double the value of turbulent fluxes (Brock et al., 2000), justifying the need to consider the larger $z_0$ values on heavily crevassed glacier areas. Additionally, the relevance of further narrowing down the model estimates range in the future depends a lot on the field of application. For large-scale, satellite-based investigations an average value between all models (e.g. 0.1 m) for heavily crevassed glacier areas might be a sufficiently accurate approximation. However, small-scale investigations on individual glaciers likely benefit from more accurate $z_0$ estimates. It is here in particular, where our study shows that UAVs are the ideal platform for investigating aerodynamic roughness length". Additionally, the $z_0$ values for smooth ice was elaborated on the grid size justification section (see minor comments 26. L279-280).

4. *Related to the above, a very nice possible outcome would be to consider a reduced-complexity parameterization of roughness for crevassed areas. This could be related to crevasse spacing (or possibly depth) or a damage factor. I suggest this because 1) crevasses can be readily mapped using high-resolution satellite imagery (Pleiades or even PlanetScope), and 2) models of glacier dynamics are inceasingly able to resolve complex stress and strain patterns near glacier termini, and this could enable an improved link to surface energy balance. It is also worth noting that unlike most glacier surfaces, the roughness of a heavily crevassed glacier terminus is not likely to see much seasonal change, as the dominant roughness elements are unaffected by snow/ice melt processes.*

    Reply: As stated on item 3. above and on item 18. (L202), the combination of satellite-based data and small-scale DEMs for reference reveals a great potential for the extrapolation

of $z_0$ estimations. However, this topic will not be discussed in more depth in the scope of this study. Therefore, the following sentence was added to the manuscript outlook: "In a next step, a combination of high-resolution DEMs from UAVs for reference $z_0$ values and satellite-based crevasse density estimates might approve valuable for future research". The authors further agree on the weaker seasonal change of heavily crevassed glaciers and addressed this issue in a way that they removed the attempt to explain some inter-annual variances by remaining snow-bridges (see minor comments item 20. L214).

**Minor comments**

1. *L7. 'best accounts' –as formulated, this sentence is a bit misleading, as you are not demonstrating that the moving-window approach is worse than the sub-grid approach (the strict interpretation of the sentence as written). Rather, your results indicate that 50m is a suitable distance for detrending, since the scale dependence for most approaches breaks down above that distance.*
   Reply: The authors agree with the referee. However, also for the moving-window approach, in the end only one $z_0$ value per 50 m x 50 m grid was provided (due to averaging the moving-window values within the grid). For clarification, the word 'best' was removed.

2. *L16. Suggest 'balance' to 'lead to'*
   Reply: Suggestion implemented.

3. *L21. I'd recommend reformulating this sentence. $z_0$ is certainly not a constant surface characteristic. At the very least it changes considerably in time (e.g. Brock et al, 2006; Smeets studies), but of course turbulence is not only affected by the surface itself, but the wind speed and direction.*
   Reply: The word constant was removed and the following sentence was added for clarification: "It is a surface characteristics and therefore independent of meteorological quantities (Lettau, 1969)".

4. *L37. I don't recall Quincey et al (2017) looking at crevasses?*
   Reply: The authors agree since Quincey et al (2017) mentioned the same issue for debris on the glacier surface, which is comparable in this matter. Nevertheless, the sentence was rephrased for clarification: "A broader-scale, heterogeneous surface topography of obstacles (e.g. crevasses) makes the definition of $z_0$ values challenging (Quincey et al., 2017)".

5. *L88. Please add some details of the final configuration here.*
   Reply: The authors understand the urge for more details here. However, we deliberately decided against adding more details in order to keep the focus on the main outcome of the manuscript and especially since the settings were discussed intensively on the work of Dachauer (2020), which is cited in the manuscript.

6. *L89. The lack of ground control points should be addressed directly here.*
   Reply: The authors agree with the referee and added the following sentence: "Due to the inaccessible glacier surface, no ground control points (GCPs) could be placed on the mapped area for georeferencing."

7. *Figure 3. Could you add depiction of the moving-window formulation (e.g. Fitzpatrick et al, 2019)? The detrending approach is indeed quite crucial for obstacle definition.*
   Reply: As described on L144, the moving-window approach is based on the work of Fitzpatrick et al. (2019). The authors decided not to explain the models in full detail since they can be looked up in the original and referenced papers. Yet, for clarification the following sentence was added: "In the raster method all sub-grids were detrended row-wise and areas below the detrended plane were neglected, assuming that they would be effectively sheltered.".

8. *L113. 'all wind directions' –by this you mean the four directions of the coordinate system defined in Fig 2b, and not every 15 degree increment, for example*
   Reply: Exactly. For clarification, the word "four" was added.

9. *L120. 'a lot' appears in the text 'a lot'. Please consider a less colloquial formulation*
Reply: 'a lot' was replaced by 'substantially'.

10. *L127. For consistency with prior descriptions, please indicate 'up-crossing' somehow here.*
Reply: Since we did not use this term in our study, we simply added the notation "(often referred to as 'zero-up-crossing' in literature)".

11. *L132.This adaptation need to be established a bit more carefully. The use of cross-profile (instead of along-profile) obstacles dates back to Lettau (1969) if not before. The rationale is that if the bumps resolved in this matter equate to the silhouette area facing the wind. This is true when surfaces do not have a clear grain, in which case you get channelized flow rather than turbulence. Note that Munro also found a 4x difference based on transect direction for ablating ice that showed a clear grain (similar to your own magnitude of differences). I actually agree with this profile rotation considering the strong grain of the surface (since you have also considering skimming flow!), but some additional justification is needed in the text.*
Reply: The authors agree with the referee and added the following sentences to further justify the adaptation: "This is because if a crevasse is aligned perpendicular to the prevailing wind direction, a wind-perpendicular transect is not able to detect the crevasse yielding to a relatively low $z_0$ value (for further explanation see Smith et al. (2016)). Such an adaptation is essential for heterogeneous and naturally streamlined roughness elements as those investigated in this study, since wind systems are influenced by the large-scale catchment topography and therefore often flow up or down the glacier (Quincey et al., 2017)". Furthermore, we referenced to Smith et al. (2016), where this adaptation is discussed intensively.

12. *L137. The precise implementation of the 'transect' approaches is not clear. Do you determine a $z_0$ value for each transect, then combine them (and how)? Or do you accumulate obstacles from all transects (as in Miles et al (2017))?*
Reply: We followed an approach where for each row/transect a $z_0$ value was determined which then were averaged to one single $z_0$ values per sub-grid. For clarification, the following sentence was added: "Thus, the final $z_0$ value for each sub-grid was then calculated by averaging the individual transect $z_0$ values within the sub-grid.".

13. *L180. Technically, these are not 'cardinal' wind directions.*
Reply: We agree that the term 'cardinal' might be misleading since the DEMs were rotated. Therefore, we decided to delete the term since the four wind directions were properly introduced in section 2.3.

14. *L194-5. Which are the 'both parameters'? Not clear.*
Reply: For clarification, the sentence was rephrased: "Locally however, both mean and median $z_0$ estimates can vary about one order of magnitude with changing wind direction."

15. *Figure 5. Please increase the font size for all axes and the colorbar.*
Reply: Suggestion implemented.

16. *Figure 6. Please use a log scale for the y-axis.*
Reply: Suggestion implemented.

17. *Figure 7. Please increase the font size for all axes and the colorbar.*
Reply: Suggestion implemented.

18. *L202. The similarity of values between glaciers (for the crevassed areas) raises an interesting question –can you parameterize $z_0$ based on crevasse density directly? If so, this would be a promising avenue to estimate $z_0$ (at least for the crevassed areas) without needing high-resolution DEMs.*
Reply: The authors appreciate the interesting input which could bring this field of research one step forward. However, we would argue that this approach would only be able to end up as a rough approximation since crevasse depth would have to be guessed from other/similar

glaciers. This is an important shortcoming because the obstacle height is the most important control parameter over the output of $z_0$ (Nield et al, 2013). Nevertheless, a combination of small-scale, high-resolution DEMs for reference values with large-scale, satellite-based crevasse density estimates might contain a valuable potential for future research and therefore was mentionend in the outlook part of the manuscript as follows: "In a next step, a combination of high-resolution DEMs from UAVs for reference $z_0$ values and satellite-based crevasse density estimates might contain a valuable potential for future research."

19. *L212. It's nice to be able to compare two different years. This similarity is also not so surprising since the roughness elements (crevasses) are probably not as transient as for other glacier surfaces.*
Reply: The UAV images and DEMs show that the roughness elements slightly change location and shape but after all we agree that the roughness elements are not as transient as for other glacier surfaces, which justifies the way this topic was mentioned in the study by not further elaborating the reasons for the inter-annual $z_0$ deviations.

20. *L214. This sentence about the decrease does not make sense –do you think the 10% reduction is meaningful? I would be very sceptical.*
Reply: The authors agree with the referee and deleted the according sentence because the deviation is so small.

21. *Figure 8. Please eliminate the duplicate colorbar.*
Reply: Suggestion implemented.

22. *Figure 9. I would again recommend plotting this with a logarithmic y-axis. Please also annotate the median obstacle sizes determined for each site (then the 50m grid size used).*
Reply: The authors acknowledge the point of the logarithmic y-axis. However, after re-evaluation we decided to stick to the non-logarithmic y-axis since both the model differences and the grid size dependence of $z_0$ values can be highlighted more effectively this way. The 50 m grid size line was included in the graph and the average obstacle sizes were annotated in the caption.

23. *L241. The nadir views at interval probably do not resolve topography very far into crevasses.*
Reply: The authors agree and that's exactly what is meant with the sentence "... the lack of reflected light from the deep crevasses."

24. *L252. I believe you are arguing that you need a DEM with high precision (rather than high accuracy). I would agree (for a microtopographic $z_0$ calculation) as measuring the local features is more crucial than the elevation of those features. The manufacturer hover accuracy estimates are not terribly relevant for your purpose, though–I would suggest that this is the best-case accuracy. Without GCPs, DEM warping can be particularly problematic.*
Reply: We agree with the referee and added the following phrase for clarification: "[...] rather a precise DEM combined with a detrending approach for the investigation of the effect of relative distances". As for the lack of GCPs, we refer to the discussion on item 1. of main comments.

25. *L256–L262. It is not clear how you measure % distortion in this context. % of what?*
Reply: For clarification, the following phrase was added: "1.7 % (horizontal length deviation in % of DEM compared to the Sentinel-2 satellite image)"

26. *L279-L280. This is a nice theoretical justification of the choice of grid size, but 50m is still quite arbitrary, looking at Figure 9, which does not show a kink but a smooth progression that might just be an artefact of the logarithmic scale of $z_0$ variability. I think you could provide better evaluation of this choice of grid size–for example, what range of $z_0$ values does this give for the smoother domains, and how does that correspond to expected values? If in fact the transition from grain to form roughness occurs at 30m (or 70m) how much are the distributed roughness estimates changed?*
Reply: The estimated $z_0$ values for each grid size has indeed been considered for the choice of the 50 m grid size. For clarification, the following text was added: "Typical $z_0$ estimates for smooth glacier ice, which for instance can be found on the upper part of Fridtjovbreen, have a

length of about 1 mm (Brock et al., 2006). The choice of a 50 m grid size can be further justified since Figure 8 shows that grid sizes below 30 m do not provide high enough values to agree with literature values.". As already discussed on Figure 9, the grid sizes above 30 m estimate similar $z_0$ values. Therefore, in this range the consideration of average obstacle sizes was important for the final choice of 50 m.

27. *L298-299. Without a doubt, katabatic winds are a predominant wind direction for mountain glaciers, and are also important for tidewater glaciers, but external forcing also plays a role, especially for the latter group. The key drivers leading to their formation (altitudinal/temperature gradients, combined with topographic chanelling) are weaker at the polar tidewater sites studied here. There are some interesting results related to this from e.g. Esau and Repina (2012). Multidirectional wind speed can play an important role in temporal variations in $z_0$, and although you don't need to go so far as to consider turbulence footprints here (Steiner et al, 2018; Nicholson and Stiperski, 2020), I think it is useful to note that the cross-and down-glacier roughness elements both influence the effective $z_0$ at a site.*
Reply: The authors appreciate the input and slightly adjusted the original text by adding the reference of Esau and Repina (2012) as follows: "However, Esau and Repina (2012) found the katabatically forced wind systems to be of less significance for tidewater glaciers, highlighting the influence of wind direction on effective $z_0$ values.". Furthermore, the last sentence of the original paragraph was removed.

28. *L305. The validation is a challenge for this study. I wonder if you can consider real-world analogues to crevassed areas (outside of glaciology) that might have been investigated previously. Lettau (1969) included a variety of other surfaces (even including urban areas) that could be relevant reference points to check the order of magnitude.*
Reply: The authors agree with the referee and compared the $z_0$ estimates with values measured above villages. Thus, the following sentence was added to the manuscript: "Outside the field of glaciology, the heavily crevassed glaciers might most effectively be compared with villages, since buildings have similar obstacle density and height. According $z_0$ values are about 0.2-0.4 m which lies within the range of estimated roughness values in this study."

29. *L397. I'd recommend making your DEMs and code publicly archived.*
Reply: We are planning to publish the DEMs in a separate publication later this year. The code will be available on the following link: `https://github.com/ArminDach/z0_UAVs`

**References:**
Brock, B. W., Willis, I. C., Sharp, M. J., and Arnold, N. S.: Modelling seasonal and spatial variations in the surface energy balance of HautGlacier d'Arolla, Switzerland, Annals of Glaciology, 31, 53–62, https://doi.org/10.3189/172756400781820183, 2000.

Brock, B. W., Willis, I. C., and Sharp, M. J.: Measurement and parameterization of aerodynamic roughness length variations at Haut Glacier d'Arolla, Switzerland, Journal of Glaciology, 52, 281–297, https://doi.org/10.3189/172756506781828746, 2006.

Chudley, T. R., Christoffersen, P., Doyle, S. H., Abellan, A., and Snooke, N.: High-accuracy UAV photogrammetry of ice sheet dynamicswith no ground control, Cryosphere, 13, 955–968, https://doi.org/10.5194/tc-13-955-2019, 2019.

Dachauer, A.: Aerodynamic Roughness Length of Crevassed Tidewater Glaciers from UAV Mapping, Master Thesis, p. 100, 2020.

Esau, I. and Repina, I.: Wind climate in kongsfjorden, svalbard, and attribution of leading wind driving mechanisms through turbulence-resolving simulations, Advances in Meteorology, 2012, https://doi.org/10.1155/2012/568454, 2012.

Fitzpatrick, N., Radi´c, V., and Menounos, B.: A multi-season investigation of glacier surface roughness lengths through in situ and remote observation, The Cryosphere, 13, 1051–1071, https://doi.org/10.5194/tc-13-1051-2019, 2019.

Lettau, H.: Note on aerodynamic roughness-parameter estimation on the basis of roughness-element description, Journal of applied meteorology, 8, 828–832, 1969.

Nield, J. M., Chiverrell, R. C., Darby, S. E., Leyland, J., Vircavs, L. H., and Jacobs, B.: Complex spatial feedbacks of tephra redistribution, ice melt and surface roughness modulate ablation on tephra covered glaciers, Earth Surface Processes and Landforms, 38, 95–102, https://doi.org/10.1002/esp.3352, 2013.

Quincey, D., Smith, M., Rounce, D., Ross, A., King, O., and Watson, C.: Evaluating morphological estimates of the aerodynamic roughness of debris covered glacier ice, Earth Surface Processes and Landforms, 42, 2541–2553, https://doi.org/10.1002/esp.4198, 2017.

Smith, M. W., Quincey, D. J., Dixon, T., Bingham, R. G., Carrivick, J. L., Irvine-Fynn, T. D. L., and Rippin, D. M.: Aerodynamic roughness of glacial ice surfaces derived from high-resolution topographic data, Journal of Geophysical Research: Earth Surface, 121, 748–766, https://doi.org/10.1002/2015JF003759, 2016.